# Variation in soil properties under different cropping and other land-use systems in Dura catchment, Northern Ethiopia

**Gebreyesus Brhane Tesfahunegn** [1]☯*, **Teklebirhan Arefaine Gebru**[2]

**1** Department of Soil Resources and Watershed Management, College of Agriculture, Aksum University, Shire-Campus, Shire, Ethiopia, **2** Department of Water Resource and Irrigation Engineering, School of Water Technology, Aksum University, Shire-Campus, Shire, Ethiopia

☯ These authors contributed equally to this work.
* gebre33@gmail.com

**Data Availability Statement:** All relevant data are within the paper and its Supporting Information file.

**Funding:** Initials of the authors who received each award: Tesfahunegn GB and Gebru TA Grant

## Abstract

There are limited reports about the impacts of long-term cropping and land-use systems (CLUS) on soil properties and nutrient stocks under smallholder farmers' conditions in developing countries. The objective of this research was to examine variation in soil properties and OC and TN stocks across the different CLUS in Dura sub-catchment, northern Ethiopia. Surveys and discussions on field history were used to identify nine CLUS, namely, tef (*Eragrostis tef* (Zucc) Trot)) mono-cropping (TM), maize *(Zea mays L.)* mono-cropping (MM), cauliflower (*Brassica oleracea var. botrytis*)-maize intercropping (IC1), red beet (*Beta Vulgaris*)-maize intercropping (IC2), cauliflower-tef-maize rotation (R1), onion (*Allium cepa* L.)-maize-onion rotation (R2), tr eated gully (TG), untreated gully (UTG), and natural forest system (NF). A total of 27 composite soil samples were randomly collected from the CLUS for soil analysis. Data were subjected to one-way analysis of variance and PCA. The lowest and highest bulk density was determined from NF (1.19 Mg m$^{-3}$) and UTG (1.77 Mg m$^{-3}$), respectively. Soil pH, EC and CEC varied significantly among the CLUS. The highest CEC (50.3 cmol$_c$ kg$^{-1}$) was reported under TG followed by NF. The highest soil OC stock (175.3 Mg C ha$^{-1}$) and TN stock (13.6 Mg C ha$^{-1}$) were found from NF. The PCA chosen soil properties explained 87% of the soil quality variability among the CLUS. Such soil properties and nutrient stocks variability among the CLUS suggested that introduction of suitable management practices that sustain the soil system of the CLUS with poor soil properties and nutrient stocks are crucial for the study area conditions.

## Introduction

Soil quality is becoming an important resource to raise crop productivity so that to meet the food required for the current and future population in developing countries as their economy mainly depends on agriculture [1–6]. Soil quality is defined as the capacity of the soil to give the intended functions for biomass and yield production [7–9]. In this study, the term soil

numbers awarde:AKU/IG/RCSD/1092/07 The full name of each funder: Aksum University Include this sentence at the end of your statement: The funders had no role in study design, data collection and analysis, decision to publish, or preparation of the manuscript.

**Competing interests:** The authors have declared that no competing interests exist.

**Abbreviations:** OC, organic carbon; TN, total nitrogen; EC, electrical conductivity; CEC, cation exchangeable capacity; Ex Na, Ex Ca, and Ex Mg, exchangeable sodium, calcium, magnesium, respectively; Pav, available phosphorus; A_h, depth of A-horizon; PCA, principal component analysis.

quality is used synonymously with soil health. Recently, however, soil quality degradation caused by inappropriate cropping system, and land-use and soil management practices, have been reported among the top development challenges that demand urgent remedial actions. Several reports have shown that soil nutrient depletion and soil physical degradation are the dominant types of degradation associated with land and soil mis-management practices in the semi-arid areas [10–15].

Soil degradation poses serious development challenges in many developing countries including Ethiopia. Soil degradation induced by land and soil mis-management systems coupled with high dependency on erratic and unreliable rain-fed farming system has aggravated the problem of food insecurity in Ethiopia. Against such problem, implementation of irrigation agriculture has been suggested as the best option in the conditions of Ethiopia [16, 17]. In support of the strategy for irrigation development, over 50 micro-dams were built-up since 1995, for being used mainly for irrigation purpose by the smallholder farmers in the Tigray region, northern Ethiopia. Using the water of the micro-dams in the region, so far a great deal of irrigation efforts has been attempted to achieve a sustainable economic, social and ecological developments. However, the sustainability of economic and ecological benefits from the micro-dams has been challenged by anthropogenic factors that increase sedimentation, soil and nutrients loadings and lowering of water use efficiency. Such factors have aggravated the rates of siltation of the micro-dams with less than 25% of their lifetime in the Tigray region [16–19].

Even though there are problems of siltation and inefficient water use, irrigation agriculture from micro-dams as water source is becoming an essential government strategy for maximizing crop production per unit land area in Ethiopia conditions [20, 21]. Irrigation is designed to increase soil water availability and thereby enhances biomass production provided that if there is little effect from salinity problem. The biomass is partly expected to be returned to the soil system to improve soil organic carbon (OC) and total nitrogen (TN) concentrations. Other practices such as reforestation of protected landscape, and agroforestry in agricultural lands have been implemented to increase soil organic matter and soil nutrients for the past three decades in Ethiopia [9, 22–24].

Many researchers (e.g., Sharma et al. [4]; Lal [25]; Mandal et al. [26]; Yesilonis et al. [27]) have also reported that planting of suitable crop types in a cropping system can play an important role in improving organic matter stock and maintaining soil nutrients such as TN stock which are important for both plants and micro-organisms. However, increasing demand for short-term production encourages farmers to cultivate continuously (mono-cropping), over-grazed fields, or removed much of the above ground biomass through fuel collection, livestock feed and building materials. Eventually, such practices reduce soil nutrients and water holding capacity, increases erosion and thereby reduce agricultural productivity. For example, comparable higher OC, TN and other soil nutrients have reported under grassland as compared to cultivated land-use type [4, 14, 25, 28, 29]. However, there are limited quantitative evidences that evaluate soil properties under different cropping systems even within the cultivated land-use type and other land-use systems managed by smallholder farmers in the northern Ethiopia conditions.

In developing countries such as Ethiopia, land has been utilized intensively for any purposes regardless of its suitability, which has resulted in severe soil quality degradation. Such degradation has explained by poor soil properties and low agricultural production [14, 26–29]. Practically, under the existing circumstances and economic conditions of farmers in developing countries such as Ethiopia there is a need to have inexpensive but environmentally sound integrated cropping system and land-use management approach to address soil quality related problems.

To maintain soil quality and reduce crop failures, intercropping which is defined as the agricultural practice of cultivating two or more crops simultaneously in the same piece of land, has also reported by many researchers (e.g., Dallal [30]; Sharaiha et al. [31]; Nursima [32]). Intercropping is practiced commonly under irrigated agriculture and sometimes in rain-fed agriculture in northern Ethiopia even though its ecological benefits over the other cropping systems such as mono-cropping, or rotation are not well documented (personal observation). The existing literatures have also shown that there is a need to understand the impacts of continuous and other types of cropping systems on soil quality indicators in order to take appropriate measures that enhance sustainable crop production. The sustainability of soils for agricultural production can be viewed using soil properties as soil quality indicators (e.g., Arshad and Martin [2]; Trivedi et al. [14]; Iqbal et al. [28]; Andrews and Carroll [33]).

Increasing crop production in Ethiopia is likely to come from agricultural intensification and diversification through irrigation and other improved soil and agronomic practices. Understanding the impacts of different cropping and land-use practices on soil properties in general and soil OC and TN stock in particular is crucial for designing sustainable soil management practices. Scientific information on site-specific soil properties is a basic tool for proper soil management in order to provide sustainable soil functions at present and in the future [2, 4, 14, 25, 34–36]. Site-specific data on soil properties could also support to dealing with spatial variability of soil nutrients and physical indicators and their influencing factors. Such information is important to formulate appropriate sustainable cropping systems and land-use type strategies [6, 12, 14, 19, 36–38].

The sustainability of crop and soil management practices to improve or maintain soil quality depends on the understanding how soils respond to different site-specific cropping and land-use practices. Soil properties as indicators of soil functions and soil quality degradation status are suggested for understanding the sustainability of soil resources [2, 14, 39, 40]. There are many reports that have generalized soil properties and soil nutrient stocks variability among different land-use types such as cultivated, grazing, grass and forest land (e.g., Trivedi et al. [14]; Chemeda et al. [15]; Wang et al. [19]; Fikadu et al. [24]; Lemenih and Hanna [41]; Batjes [42]; Yimer et al. [43]). However, there have been limited studies about the impacts of site-specific long-term irrigated and non-irrigated cropping and land-use systems under different soil and crop management practices on soil properties and soil nutrient stocks for smallholder farmers' conditions in developing countries.

Since the roles of most agricultural practices are site-specific, the same management strategy cannot be recommended using the existing reports for the conditions of smallholder farmers in northern Ethiopia. Thus, it necessitates knowing the extent of soil quality degradation in terms of soil physical and chemical properties and nutrient stocks under different management practices such as irrigation and rain-fed cropping and other land-use systems and soil management practices. There is also insufficient information about which soil properties (indicators) to be monitored over time with regard to the effects of cropping systems and other land-use practices in the study area conditions [5, 12, 19, 38]. This study was thus hypothesized to test that there is significant variability in soil properties and soil OC and TN stocks across the different cropping and other land-use systems. The objectives of this research were to: (i) examine variation in soil properties under different long-term cropping and land-use systems (CLUS); (ii) evaluate soil organic carbon and total nitrogen stocks across the different CLUS; and (iii) examine soil properties that explain better for soil quality variability across the different CLUS in the conditions of similar to the Dura sub-catchment, northern Ethiopia.

## Materials and methods

### Study area

This research was carried-out from February 2015 to June 2015 in the Dura sub-catchment of Tigray region, northern Ethiopia (Fig 1). The area of the Dura catchment is 5000 ha and that of Dura sub-catchment is 1240 ha. Altitude of the sub-catchment ranges between 2050 and 2650 m above sea level [44]. In the study sub-catchment, mean annual temperature of 22°C and rainfall of 700 mm were reported using 35 years of meteorological data. The study sub-catchment receives normal rainfall during June to early September which is unimodal (Meteorology Agency-Mekelle branch). Crop and livestock mixed-farming is commonly practiced. However, crop production is the dominant farming system for farmers' livelihoods. Arable land is dominated over the other land-use types in the study sub-catchment.

In the Dura sub-catchment, both rain-fed and irrigation agriculture have been practiced for more than 2 decades. But rain-fed agriculture which is the oldest practice is still dominating in area coverage over irrigated land. About 100 ha farmland has been irrigated since 1996 in the Dura sub-catchment. Afforested area, pasture, scattered woody trees, bushes and shrub lands were also found in the study sub-catchment. The dominant soils in the study sub-catchment includes: Eutric Cambisols on the steep slope, Chromic Cambisols on the middle to steep slopes and Chromic Vertisols on the flat areas [45]. This study sub-catchment was selected as it represents the mid-highland agro-ecology conditions having different CLUS with practices such as irrigated and non-irrigated fields and diversified soil management experiences under the smallholder farmers in the northern Ethiopia.

### Identification of long-term different cropping and land-use systems

Reconnaissance surveys coupled with formal and informal group discussions with farmers and development agents (DAs) were used to identify the different irrigated and non-irrigated cropping and land-use systems (CLUS) and their specific soil management practices in the Dura sub-catchment, northern Ethiopia. During the field surveys in February and March 2015 the researcher together with the DAs visited the study catchment to get an overall impression

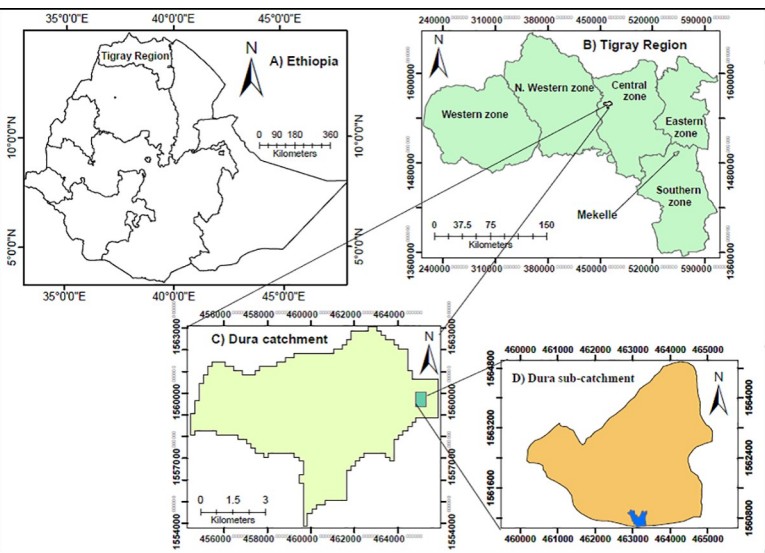

**Fig 1.** Location of the study area: (A), Ethiopia (B)Tigray Region, and (C) Dura catchment.

about the irrigation command area (an area where irrigated agriculture is practiced), adjacent rain-fed cropping and other land-use systems. The purpose of the reconnaissance survey was to characterize the fields' historical cropping system, soil management, agronomic practices and field features. During the survey, participatory tools such as field observation, transect-walks and group discussions were employed. The transect-walks were done twice, that is, from the east to the west and also from the north to south direction of the study sub-catchment in order to observe different cropping and other land-use systems. This was done by the team composed of two researchers, three (3) DAs and randomly selected 10 farmers from the study sub-catchment.

Three group discussions sessions were held in order to reach consensus among the participants about the descriptions of the irrigated and non-irrigated fields that were selected during the transect-walks. On the basis of the farmers' views in the group discussion final consensus, nine (9) dominant cropping and land-use systems (CLUS) were identified and described all their corresponding crop and soil related management practices (Table 1). In addition, the

**Table 1. Different cropping and land-use systems identified in the Dura sub-catchment, northern Ethiopia.**

| S. no. | Cropping and other land-use system | Description |
|---|---|---|
| 1 | [a]Tef (*Eragrostis tef* (Zucc) Trot*)*) mono-cropping (TM) | Continuous tef crop has been grown (mono-cropped) for more than 18-years at the same field. Inorganic fertilizer of 100 kg ha[-1] DAP and 50 kg ha[-1] urea applied in each cropping seasons annually, but recently the application was 100 kg ha[-1] of each of these fertilizers. Tillage frequency to prepare seed bed ranges between three and six times, depending on field soil conditions and farmers resources availability. The tef fields were rain-fed and located near to the irrigated fields under the same soil type and geology, but far from homesteads. Soil and water conservation (SWC) practices observed only at the boarder that separates between two fields owned by different farmers. No manure or compost fertilizer applied. Soil samples were collected just after harvested the tef crop. |
| 2 | Maize (*Zea mays L.*) mono-cropping (MM) | Continuous maize crop has been grown for more than 30-years at the same field. Fields are located just at homesteads and received about 5 to 8 tones ha[-1] of manure each year. Sometimes, 100 kg DAP ha[-1] and 50 kg urea ha[-1] were applied annually if organic sources are small or unavailable. Tillage frequency is at most three times for maize cultivated land. This is a rain-fed cropping system, and relatively it has intensive SWC practices. Soil samples were collected just after harvested the maize crop. |
| 3 | Cauliflower (*Brassica oleracea var. botrytis*) with maize intercropping (IC1) | Intercropping of cauliflower with maze was practiced during the irrigation period for a consecutive of four years. Flood irrigation for land preparation and furrow irrigation just after planting was used. Maize was planted two weeks after cauliflower. Soil samples were collected at maturity stage of both crops. During the rain-fed season tef intercropped with "*Nuhig*" (*Guizotia abyssinica L.*) was used for crop rotation. Fertilizer rate of 100 kg DAP ha[-1] and 50 kg urea ha[-1] were applied to all cropping seasons (twice a year). The land was ploughed three times. SWC practices were found at field boarders. Small rate of manure/compost (1 ton ha[-1]) was applied occasionally just at planting of cauliflower. |
| 4 | Red beet (*Beta Vulgaris*) with maize intercropping (IC2) | For land preparation, flood irrigation was used. Intercropping of red beet with maize was practiced using furrow irrigation for a consecutive of five years. Maize was planted three weeks after red beet. Soil samples were collected at maturity stage of both crops. During the rain-fed season tef intercropped with "*Nuhig*" (*Guizotia abyssinica L.*) was used to rotate the system. Fertilizer rate of 100 kg DAP ha[-1] and 50 kg urea ha[-1] were applied during the cropping seasons (twice a year). The land was ploughed three times. A small rate of manure/compost (1 ton ha[-1]) was applied occasionally just at planting of red beet. |

*(Continued)*

**Table 1.** (Continued)

| S. no. | Cropping and other land-use system | Description |
|---|---|---|
| 5 | Cauliflower - tef - maize rotation (R1) | Irrigated (furrow irrigation) sole cauliflower was first planted. After this crop harvested, rain-fed tef was planted. After tef, irrigated maize was planted and then the rotation was continued to cauliflower followed by maize again for two terms (6 years). Soil samples were taken during the maturity stage of irrigated maize crop at the end of term two. The fertilizer rate applied for all of the crops during the rotation was 100 kg DAP and 50 kg urea ha$^{-1}$. A small rate of manure/compost (1 ton ha$^{-1}$) was applied occasionally just at planting time of cauliflower and maize. |
| 6 | Onion (*Allium cepa* L.) - maize - onion rotation (R2) | Irrigated (furrow irrigation) onion was first planted. After this crop, rain-fed maize was planted as a rotational crop. After maize, irrigated onion was planted again and then continue the rotation to rain-fed maize and back to onion irrigation for two terms (6 years). Soil samples were taken during maturity stage of the irrigated onion at the end of the term two. The fertilizer rate applied to both crops included 100 kg DAP ha$^{-1}$ and 50 kg urea ha$^{-1}$. A small rate of manure/compost (1 ton ha$^{-1}$) was applied occasionally just at planting of irrigated onion. |
| 7 | Treated gully (TG) | The gully in the irrigation command area was treated using Sesbania (*Sesbania sesban*) and Leucena (*Leuceana leucacephala*) legume trees which established 20-years ago. Naturally regenerated grasses have also grown well on the bed and sides of the gully and have used only by cut and carrying system. The gully treatment has entirely dependent on biological SWC. Excess irrigation water from the fields was drained to the treated gully. |
| 8 | Untreated gully (UTG) | The untreated gully had no improved management practices, e.g., no SWC, no enrichment of tree, shrub and grass species. This land has not been contributed to the local community livelihood for many years. According to local farmers, the estimated age of gully is more than 100 years. |
| 9 | Natural forest land system (NF) | This is less disturbed land which is used as a reference. NF has native trees, vegetation and grass cover. Example of dominant tree species include: *Acacia etbaica*, *Acacia abyssinica*, *Olea europaea*, *Acacia lahai*, *Dodonaea euquistifolia*, *Dovyelis abyssinica;* and grass species such as *Datura stramonium L.*, *Cynodon dactylon (L.) pers*, *Trifollium rueppellianum Fresen.* |

DAP, Di ammonium phosphate

[a]Tef followed by maize is the dominant crop in the study catchment of northern Ethiopia. Tef is an annual cereal crop (belonging to the grass family) with very fine seeds that requires field with fine seedbeds.

CLUS were geo-referenced and described their topographic features (Table 2). Elevations and slope of the CLUS considered in this study vary between 2040 and 2105 m; and 1.3 and 2.5%, respectively. The CLUS were located between Universal Transverse Mercator 37 North (UTM-37N) of 461736 and 463657 North latitude, and 1559196 and 1561482 East longitude. Such fields were selected because the soil and crop specific management practices perhaps affect the sustainability of natural resources, crop productivity and soil fertility utilization in the sub-catchment. The selected CLUS were located on Chromic Vertisols adjacent to each other at a distance that ranges between 50 and 150 m which covered a total land area of 15 ha in the study sub-catchment.

Generally, the parent material of the sampling fields is basic metavolcanics. The implication is that any variability in soil properties could be induced due to irrigation, cropping type and

soil management practices in the study sub-catchment. For instance, it was observed in the field that flood and furrow irrigation caused fine soil particles to detach and transport from the source site and deposited in the depositional areas. Differences in slopes even slight differences can aggravate soil particles transportation by flowing water due to furrow irrigation. The same is true for rainfall generated runoff effect on soil particles transportation and deposition. Such practices are the derivers for soil textural variability across the CLUS in the study sub-catchment.

From the total nine (9) CLUS identified, four (4) were from irrigated fields, two (2) from rain-fed cropping system, and three (3) from other land-use systems. The cropping and land-use systems (fields) selected were: (i) Tef (*Eragrostic tef* (Zucc) Trot*)) mono-cropping (TM), (ii) Maize (*Zea mays L.)* mono-cropping (MM), (iii) Cauliflower (*Brassica oleracea* var. botry-tis) with maize intercropping (IC1), (iv) Red beet (*Beta Vulgaris*) with maize intercropping (IC2), (v) Cauliflower - tef - maize rotation (R1), (vi) Onion (*Allium cepa* L.) - maize - onion rotation (R2), (vii) Treated gully (TG), (viii) Untreated gully (UTG), and (ix) Natural forest land-use system (NF) (Table 1). The first two land-use systems (TM and MM) were selected from the rain-fed fields adjacent to the irrigation command area, whereas IC1, IC2, R1 and R2 were selected from the irrigated crop fields. TG and UTG were also found within the irrigation command area. An adjacent natural forest land-use system (NF) was used as a reference while compared with the impact of irrigation and non-irrigation cropping and land-use systems on soil properties, and carbon and nitrogen stocks.

## Sample designing, soil sampling and analysis

The targeted fields (population) were all the cropping and land-use systems (CLUS) practiced in the Dura sub-catchment. In consultation with farmers nine CLUS that dominantly available in the sub-catchment were selected. The soil samples were taken from the nine irrigated and non-irrigated CLUS which were selected from the sub-catchment. Composite soil samples were collected from the selected sampling points in each CLUS using judgmental sampling on the basis of reliable historical and physical knowledge of experts and local farmers as described in Table 1. Soil samples were collected in May 2015. Three sampling units replicated in the nine CLUS were identified by experts' judgment on the basis of field homogeneity. Identification of the sampling units using expert knowledge is very efficient as it is quick and easy to select the sampling units.

Considering the costs of soil analysis and its statistical representativeness a total of 27 composite soil samples from the three sampling units (9 CLUS x 3 sampling units) were collected. The sampling units were considered as replication of soil sampling from the CLUS. The soil sampling unit plot area was 48 m$^2$ (6 m x 8 m). The soil sampling plots land features are described in Table 2. In each sampling unit plot 10 pairs of randomly selected coordinate points were identified. From the 10 geo-referenced points in each sampling plot, three sampling points were selected using simple random sampling technique whereby the composite soil sample from each plot was collected.

The soil samples were taken from each sampling point at 0-20 cm soil depth. This sampling depth was selected as it is where most soil changes are occurred due to long-term cropping systems, land-use types, and soil and water management practices including irrigation agriculture. Three soil samples were collected from each sampling unit in a plot and pooled (composited) into a bucket and mixed thoroughly to form a single homogenized sample. A sub-sample of 500 g soil that represented the pooled sample in the bucket was taken from each sampling unit plot, and air dried and sieved through 2 mm mesh sieves. In addition, three undisturbed soil samples were collected from each CLUS sampling unit plots at 0-20 cm soil

**Table 2. Topographic features of each soil sampling unit selected from the different cropping and land-use systems in the Dura sub-catchment, northern Ethiopia.**

| Cropping and land-use system | Sampling point 1 | | | | Sampling point 2 | | | | Sampling point 3 | | | |
|---|---|---|---|---|---|---|---|---|---|---|---|---|
| | Elevation (m) | Slope (%) GMP | UTMª | | Elevation (m) | Slope (%) GMP | UTM | | Elevation (m) | Slope (%) GMP | UTM | |
| | | | Latitude | Longitude | | | Latitude | Longitude | | | Latitude | Longitude |
| M | 2066 | 2.0 | 462488 | 1559941 | 2064 | 2.5 | 462412 | 1559951 | 2065 | 2.0 | 462468 | 1559946 |
| MM | 2067 | 2.5 | 461736 | 1559812 | 2080 | 3 | 461675 | 1559852 | 2081 | 2.5 | 461581 | 1559861 |
| IC1 | 2072 | 2.5 | 461924 | 1559672 | 2073 | 2.0 | 461902 | 1559679 | 2068 | 1.5 | 461853 | 1559780 |
| IC2 | 2039 | 1.3 | 463409 | 1559234 | 2044 | 1.5 | 463411 | 1559229 | 2055 | 2.0 | 463437 | 1559227 |
| R1 | 2065 | 2.0 | 461903 | 1559794 | 2042 | 1.4 | 463548 | 1559073 | 2040 | 1.5 | 463579 | 1559028 |
| R2 | 2040 | 1.5 | 463015 | 1559479 | 2047 | 1.5 | 463053 | 1559460 | 2048 | 1.5 | 463054 | 1559445 |
| TG | 2042 | 1.5 | 462355 | 1559196 | 2045 | 1.5 | 463554 | 1559211 | 2044 | 1.5 | 463540 | 1559204 |
| UTG | 2047 | 1.5 | 462856 | 1559450 | 2049 | 1.5 | 462855 | 1559477 | 2049 | 1.5 | 462863 | 1559462 |
| NF | 2105 | 2.5 | 463657 | 1561482 | 2110 | 3.0 | 463,856 | 1561592 | 2103 | 2.0 | 464039 | 1561523 |

TM, Tef *(Eragrostis tef* (Zucc) Trot*))* mono-cropping; MM, Maize *(Zea mays L.)* mono-cropping; IC1; Cauliflower (Brassica oleracea var. botrytis) with maize intercropping; IC2, Red beet *(Beta Vulgaris)* with maize intercropping; R1, Cauliflower-tef-maize rotation; R2, Onion *(Allium cepa* L.)- maize - onion rotation; TG, Treated gully; UTG, Untreated gully; NF, Natural forest land-system.

ªUniversal Transverse Mercator 37 North (UTM-37N) in meters is the projection system.

depth using 5·0 cm long by 5·0 cm diameter cylindrical metal core sampler to determine soil dry bulk density.

The analysis of the soil samples was carried-out following the standard laboratory procedures in JIJE Analytical Testing Service Laboratory in Addis Ababa, Ethiopia. Soil texture was determined using the Bouyoucos hydrometer method [46] and soil dry bulk density (DBD) by the core method [47]. Total porosity was calculated from the DBD and assumed average particle density (PD) of 2.65 Mg m$^{-3}$ as (1-DBD/PD) × 100 [48]. The depth of A-horizon was directly measured as an average of the three pits opened (0·60 m depth) in each sampling unit plot.

Soil pH was determined in 1:2·5 soil to water ratio using pH meter combined glass electrode [49], electrical conductivity (EC) in 1:2·5 soil to water ratio using conductivity meter [50], soil organic carbon (OC) by the Walkley-Black method [51], available phosphorus (Pav) by Olsen method [52], and total nitrogen (TN) by Kjeldhal digestion method followed by distillation and titration [53]. Cation exchange capacity (CEC) was determined by ammonium acetate extraction buffered at pH 7 using flame photometer [54].

Exchangeable bases (Ca$^{2+}$, Mg$^{2+}$, K$^+$ and Na$^+$) were analyzed after extracted in a 1:10 soil/solution ratio using 1M ammonium acetate at pH 7.0. Readings for Ca$^{2+}$ and Mg$^{2+}$ in the extracts were analyzed using atomic absorption spectrophotometer whereas Na$^+$ and K$^+$ were determined by flame photometry [55].

## Derived soil parameters

Soil structural stability index of the different CLUS is estimated [56, 57] as:

$$SSSI = \frac{1.724 SOC(\%)}{clay(\%) + silt(\%)} \times 100 \tag{1}$$

Where, SSSI (%) is soil structural stability index, SOC is soil organic carbon of each CLUS, and clay + silt is combined clay and silt content determined from each CLUS. SSSI < 5% indicates structurally degraded soil; 5% < SSSI < 7% indicates high risk of soil structural degradation; 7% < SSSI < 9% indicates low risk of soil structural degradation; and SSSI > 9% indicates

sufficient SOC to maintain the structural stability. A higher the SSSI value, a better would be in maintaining soil structural degradation [56, 57]. The implication of Eq (1) is that optimum ratio (proportion) between the contents of soil organic matter and fine soil particles in the soil system are crucial to maintain the soil structural stability, in which it expresses the risk for soil structural degradation associated with SOC depletion [56, 57]. Structural stability is one of the indices of stable soil aggregates and thus the higher the structural stability index values better will be the in OC and thereby in soil structure which could encourage soil resistance to disruptive forces [56].

Base saturation percentage was calculated by divided the sum of base forming cations ($Ca^{2+}$, $Mg^{2+}$, $K^+$ and $Na^+$) by CEC and then multiplied by 100%. Exchangeable sodium percentage (ESP) was calculated by divided exchangeable $Na^+$ by CEC. The ESP threshold of 15% was used to classify sodium hazard, that is, sodic soils are those with ESP of more than 15%. Sodium adsorption ratio (SAR) for each CLUS was calculated [58–60] as:

$$SAR = \frac{Na^+}{\sqrt{\frac{(Ca^{++}+Mg^{++})}{2}}} \tag{2}$$

Where, SAR is sodium adsorption ratio ($cmol\ kg^{-1})^{0.5}$; and $Na^+$, $Mg^{2+}$ and $Ca^{2+}$ are exchangeable sodium, magnesium and calcium, respectively, in $cmol_c\ kg^{-1}$. SAR < 12 indicate non sodicity and values >12 indicate sodic soils [58].

The relationship between soil OC and TN as represented by the ratio of C to TN was derived as an indicator of soil quality status. The C:N ratio is a sensitive indicator of soil quality when assesses soil carbon and nitrogen nutrient balance. It is used as a sign of soil nitrogen mineralization capacity [61, 62]. A high C:N ratio indicates the slowdown decomposition rate of organic matter by limiting soil microbial activity. On the other hand, low ratio of C:N ratio could show the accelerated process of microbial decomposition on organic matter and nitrogen, in which this is not conducive for carbon sequestration [61–63].

Soil OC and TN stocks ($Mg\ ha^{-1}$) for each CLUS were calculated using equivalent soil mass approach and elemental mass per area (Eq 4) described by Ellert and Bettany [64] as:

$$M_{soil} = \rho_b \times T \times 10000\ m^2\ ha^{-1} \tag{3}$$

$$OC(or\ TN)Stock = Conc \times \rho_b \times T \times 10000\ m^2\ ha^{-1} \times 0.001\ Mg\ kg^{-1} \tag{4}$$

Where: $M_{soil}$ is soil mass per unit area ($Mg\ ha^{-1}$), $\rho_b$ is dry bulk density ($Mg\ m^{-3}$), $T$ is thickness of soil layer (m), *OC (or TN) Stock* is elemental mass per unit area of soil organic carbon or total nitrogen stock ($Mg\ ha^{-1}$), and *Conc* is elemental soil organic carbon or total nitrogen concentration ($kg\ Mg^{-1}$). The equivalent soil mass has reported to be the most appropriate method for assessing N and C stocks because in the first case, such stocks are not influenced by changes in soil bulk density and in the second case, even if the stocks have been slightly modified because of changes in bulk density, the stocks are more sensitive to changes in land use [63, 64]. The OC and TN stocks were quantified consistently using the same equivalent soil mass as the basis for comparison. Because the source of the error is reported as OC and TN stocks are being compared at fixed depth that contains different soil masses. To be able to compare the same soil mass and avoid interference of bulk density changes with OC and TN changes, such stocks was calculated using the additional thickness required to attain equivalent

soil mass as described by Ellert and Bettany [64]:

$$T_{add} = \frac{(M_{soil,equiv} - M_{soil,surf})0.0001 \; ha \; m^{-2}}{\rho_{b \; subsurface}} \quad (5)$$

Where, $T_{add}$ is additional thickness of subsurface layer required to attain the equivalent soil mass (m), $M_{soil,equiv}$ is equivalent soil mass which is the mass of heavies horizon (Mg ha$^{-1}$), $M_{soil,surf}$ is soil mass in surface layer or genetic horizon (Mg ha$^{-1}$) calculated using the adjusted soil thickness (Eq 6) and $\rho_{b \; subsurface}$ is bulk density of subsurface layer (Mg m$^{-3}$). The untreated gully (UG) land use system with soil mass of 3520 Mg ha$^{-1}$ wass designated as the equivalent soil mass and the thickness required to attain 3520 Mg ha$^{-1}$ by each of the other CLUS profile was used to calculate the quantity of the corresponding equivalent soil mass.

The soil depth (T) of the other CLUS that contain the same mass of soil as the corresponding layer in the reference profile (natural forest, NF) was calculated. The genetic thickness (T) of the soils of the other CLUS such as cultivated soils was corrected (*T corrected)* using Eq (6), assuming that the bulk density and depth of such CLUS soils was originally the same as those of the corresponding forest soils [65] as:

$$T \; corrected = (\frac{\rho_{b \; forest}}{\rho_{b \; other \; CLUS}})T \quad (6)$$

Where, *T corrected* is the adjusted depth for the profile/layer with respect to the reference soil (NF), T is the soil thickness of each of the CLUS considered for evaluation with respect to NF, $\rho_{b \; forest}$ is bulk density of natural forest and $\rho_{b \; other \; CLUS}$ is bulk density of each of the other CLUS included in this study. Using the *T corrected* and the corresponding bulk density, soil mass of the genetic horizon (surface) wad calculated. In addition, the soil thickness that corresponded to the equivalent soil mass of all the CLUS was used to calculate the OC and TN stocks. The mass of elements per unit area in an equivalent soil mass was calculated as a summation of equivalent soil mass of the surface soil (genetic horizon) and those in the additional thickness of subsurface layer [64].

Natural forest (less disturbed system) was used as a reference while assessed the amount of soil OC and TN stock reduction due to the effects of each CLUS. Thus, the difference between NF and any CLUS divided by NF and multiplied by 100% was used to show the reduction of OC and TN from the soil stocks in the different CLUS. Reduction in soil OC and TN implies that there is contribution towards increasing global greenhouse gases that emitted to the atmosphere.

## Data analysis

Data were analyzed using statistical software package of SPSS 20·0, SPSS Inc. International Business Machines Company, Chicago, USA. One-way analysis of variance was carried out to test the mean differences of the soil properties among the nine cropping and land-use systems. Data were tested for the assumption of normal distribution. Means were separated using Least Significant Differences at probability level (P) $\leq$ 0·05. Data were also subjected to descriptive, correlation (r) and factor analysis.

Correlations among the soil properties were checked by Pearson product moment correlation test (two-tailed) at P $\leq$ 0·05. The principal component analysis (PCA) was also used to extract factor components and reduce variable redundancy. The PCA was thus used to examine the relationship among the 22 soil properties by statistically grouped into five principal components (PCs) using the Varimax rotation procedure. The five PCs with eigenvalues > 1 that explained at least 5% of the variation of the soil properties with respect to the cropping

and land-use systems were considered. Varimax rotation with Kaiser Normalization resulted in a factor pattern that highly loads into one factor [66]. If the highly weighted variables within PC correlated at the correlation (r) value < 0.60, all variables were retained in the PC. Among the well-correlated variables (r ≥ 0.60) within PC, a variable with the highest partial correlation coefficient and factor loading was retained in the component factor. Partial coefficient analysis was used to detect any multi-collinearity effect among the variables in a given PC.

Note that only variables with factor loadings > 0.7 were retained in the PC. If the loading coefficient of a variable was > 0.7 in more than one component, it was suggested to select from the component holding with the highest coefficient for that variable [67]. Communalities that estimate the portion of variance of each soil parameter in the component factor was also considered while selected a variable to be retained in the PC. A higher communality for a soil parameter indicates a higher proportion of the variance is explained the component factor by the variable. Less importance should be ascribed to soil parameters with low communalities when interpreting the PC factors [66, 67].

### Ethics approval

Ethical approval was obtained from the Research and Community Services Directorate Director Research Ethics Review Committee of Aksum University, Ethiopia to conduct this study. Full right was given to the study participants to refuse and withdraw from participation at any time. Confidentiality of respondents was preserved by the researchers during data collection of soil samples and soil and crop management history. It was also noted that this research has no any activities that directly related to human being as it is directly related to the physical environment. It is a norm to access and take soil samples from the sampling sites after introducing about the objective of the research and showing the approved proposal by Aksum University (Ethiopia) for village development agents' (DAs). The DAs were the focal person for Bureau of Agriculture and so directly communicate with farmers on issues related to this research and also while technology dissemination. Soil samples were collected with the oral permission of the farmers.

## Results ad discussion

### Effects on soil physical properties

The soil physical properties significantly varied among most cropping and land-use systems (CLUS) in the Dura sub-catchment, northern Ethiopia. There were significant differences in clay, silt, sand, bulk density, porosity, soil structural stability index, and A-horizon among most of the CLUS (Table 3). The soil clay contents of the CLUS varied significantly between 26 to 74%, with the lowest and the highest values observed from TM and R2, respectively.

The lowest silt (22.7%) and sand (3.7%) contents were observed from R2 whereas the highest silt (43%) from NF and sand (39.0%) from TM were observed. The highest sand content in the TM may be associated with repeated cultivation using the long-term inorganic fertilization in which such practices aggravate effect of erosion that erode fine soil particles and leaves coarser particles [68– 70]. Long-time flood and furrow irrigation can also increase soil particles displacement and transportation in the irrigation fields in which this can influence the variation in soil properties in the irrigation fields.

In addition, the severe erosion during the rain-fed season also contributes for soil textural variability through the deposition-transportation process. The high clay component for the R1 and R2 systems are probably due to being in a more level depositional or alluvial landscape and may represent the Chromic Vertisols. The mean clay (44.2%), silt (33.2%) and sand (22.6%) contents of all the CLUS indicated that the sub-catchment soil has dominated by clay

**Table 3. Mean soil physical properties variability among the different cropping and land-use systems at 0-20 cm depth in the Dura sub-catchment, northern Ethiopia.**

| Physical soil property | Cropping and land-use system (CLUS) | | | | | | | | | Mean |
|---|---|---|---|---|---|---|---|---|---|---|
| | TM | MM | IC1 | IC2 | R1 | R2 | TG | UTG | NF | |
| Clay (%) | 26.3f | 32.0e | 43.7d | 48.7c | 54.0b | 73.7a | 49.7c | 36.7e | 33.7e | 44.2 |
| Silt (%) | 33.4d | 39.0b | 36.7c | 30.0e | 33.7d | 22.7f | 34.7cd | 25.3f | 43.3a | 33.3 |
| Sand (%) | 40.3a | 29.0b | 19.7d | 21.3d | 12.3e | 3.7f | 15.7e | 38.0a | 23.0c | 22.6 |
| DBD (Mg m$^{-3}$) | 1.59b | 1.39e | 1.47d | 1.49cd | 1.51c | 1.50c | 1.30f | 1.77a | 1.19h | 1.36 |
| TP (%) | 40.1g | 47.5c | 44.5ef | 43.4f | 46.2d | 45.3de | 50.3b | 33.2h | 55.1a | 45.1 |
| SSSI (%) | 1.85f | 3.93c | 2.68d | 2.67d | 2.218e | 1.99e | 7.17b | 1.15g | 12.1a | 3.89 |
| A-h (m) | 0.104d | 0.138c | 0.110d | 0.102d | 0.105d | 0.106d | 0.187b | 0.017e | 0.195a | 0.120 |

Means followed by different letters in the same row are significantly different at probability level (P) = 0·05.

DBD, dry bulk density; TP, total porosity; SSSI, soil structural stability index; A-h, A-horizon depth; TM, Tef*(Eragrostis tef* (Zucc) Trot*)* mono-cropping system; MM, Maize *(Zea mays L.)* mono-cropping system; IC1; Cauliflower (*Brassica oleracea* var. botrytis) with maize intercropping; IC2, Red beet (*Beta Vulgaris*) with maize intercropping; R1, Cauliflower - tef - maize rotation; R2, Onion (*Allium cepa* L.) - maize - onion rotation; TG, Treated gully; UTG, Untreated gully; NF, Natural forest.

followed by silt soil texture. Fields with higher clay content such as R2 are considered by local farmers as difficult for workability and so susceptible to the problem of water logging in which this is in agreement with the reports reported by Barrios and Trejo [68]; Mairura et al. [69] and Tesfahunegn et al. [70].

On the other hand, there were non-significant differences in soil sand contents among some of the CLUS, e.g., between TM and UTG, IC1 and IC2, R1 and TG (Table 3). This could be attributed to the fact that sand texture is the soil property that does influence little by some of the CLUS and their activities and so by erosion-deposition processes. The present finding on soil sand content is consistent with Shepherd et al. [71] who reported that there is no significant effect of land-use systems on soil particle size distribution. However, such reports contradicted with that of Kauffmann et al. [72]; Voundi Nkana and Tonye [73] and Agoumé and Birang [74] who reported that continuous cropping and intensive land-use systems have significantly affected soil particle size distribution. Such discrepancy in soil particles could be attributed to the duration of the cropping system, variability in management practices and weather conditions, and effects of variation in topography.

The lowest dry bulk density (DBD) was recorded from NF (1.19 Mg m$^{-3}$) followed by TG (1.32 Mg m$^{-3}$) and MM (1.39 Mg m$^{-3}$). Conversely, the highest bulk density was found from UTG (1.77 Mg m$^{-3}$) followed by TM (1.59 Mg m$^{-3}$) and IC2 (1.50 Mg m$^{-3}$). However, there were non-significant differences in DBD between IC1 and IC2, and R1 and R2. Generally, DBD was found to be higher in mono-cropping than intercropping and rotation cropping systems; and in intercropping than crop rotation fields (Table 3). The exceptional lower DBD from MM could be associated with the effect of manure and compost on soil system whereby farmers regularly applied to maize fields in each cropping season. The DBD of NF, TG and MM were found within the ideal critical levels (1·00-1·40 Mg m$^{-3}$) as described as an ideal soil condition for plant root growth and water holding capacity by Arshad et al. [75]. However, the other cropping and land-use systems considered in this study showed DBD values higher than the critical level in which this implies the need for adopting appropriate practices that improve soil bulk density.

Total porosity, SSSI and A-horizon values were significantly varied among the CLUS, with the highest of these parameters recorded from NF and the lowest from UTG (Table 3). The trend of these parameters is similar to that of DBD but in the opposite direction. The variation in total porosity, SSSI and A-horizon among the different CLUS could be attributed to the

differences in soil organic matter (SOM) contents. In fact, land-use systems such as NF, TG and MM which have received higher OM sources can improve the quality of soil properties. Soils with a good physical quality have a stable structure which resists for the effects of erosion [74, 75]. The risk of soil structural degradation associated with SOC depletion was found to be higher in UTG followed by TM even though R1, R2, IC1, IC2 and MM are also showed structurally degraded soil with SSSI < 5%.

Consistent with the present finding, substantial reports have shown that degraded soil that receives higher SOM can improve soil porosity, soil structure and depth of A-horizon. Improving such soil attributes would enhance soil water-holding capacity, decreases runoff and soil losses and eventually increases agricultural production (e.g., Agoumé and Birang [74]; Arshad et al. [75]; Sojka and Upchurch [76]; Evanylo and McGuinn [77]; Sally and Karle [78]; Moghadam et al. [79]). Existing reports have also indicated that soils dominated by silt and clay fractions are impervious, impermeable, puddled and are subjected to alternate drying and wetting cycles which do not promote stability of aggregates [77, 78]. In addition, cultivated fields treated with mineral fertilizer consecutively for many years such as TM and irrigation fields (e.g., IC1, IC2) showed relatively poor soil physical properties because such cropping systems were treated using only mineral fertilizer for a long-term that deteriorates soil physical properties. However, the trends of rates of mineral fertilizer have been increased from time to time in Ethiopia. The present result of structural physical degraded soils could be associated with the continuous application of chemical fertilizer for more than 2 decades which in line with the reports from Moghadam et al. [79] and Ayoola [80] who reported a negative effect of continuous usage of mineral fertilizer on the soil system.

### Effects on soil chemical properties

**Effects on soil pH and electrical conductivity.** There were statistically significant differences in soil chemical properties among most cropping and land-use systems (CLUS) in the study sub-catchment (Table 4). The soil pH varied significantly from 6.94 in TM to 8.50 in R1. The higher pH in R1 could be associated with the quick natural weathering of rocks and soil processes that may result in much dissolved salts in the soil system coupled with low leaching of basic cations from soil exchange complex. However, to give specific reason for having a higher pH in R1 this merits further investigation. There was also non-significant differences in soil pH among some of the CLUS (e.g., between MM and NF; and among IC1, IC2, and TG). The mean soil pH (7.68) of all the CLUS indicates that the study catchment soil is categorized as moderately alkaline in reference to the classification for African soils reported by Landon [81]. Generally, the CLUS in the catchment showed that soil pH values are within the critical levels (6.5-8.5) reported in literature (e.g., Sanchez et al. [82]; Tesfahunegn et al. [83]. The present pH values indicate that soil pH is not a key problem to monitor effects of the different cropping and land-use systems on soil quality indicators.

The highest soil EC was recorded from the irrigated fields of IC2 (0.510 ds m$^{-1}$) followed by R2 (0.390 ds m$^{-1}$) whereas the lowest was found from rain-fed field of TM (0.057 ds m$^{-1}$). However, there were non-significant differences in EC among many of the CLUS (e.g., MM, IC1, R1, R2, TG, UTG and NF) (Table 4). According to Landon [81], soil EC determined from the different CLUS is categorized as non-saline even though EC was found to be higher in the irrigation fields as compared to the other CLUS. The most likely reason for having low EC even in the irrigated fields could be attributed to the acceptable irrigation water quality, and irrigation method (furrow), and heavy rainfall during the summer season (June to early September) which may contribute for timely leaching of salts from the root zone. It is thus suggested to

**Table 4. Mean soil chemical properties variability among the different cropping and land-use systems at 0-20 cm depth in the Dura sub-catchment, northern Ethiopia.**

| Chemical soil property | Cropping and land-use system (CLUS) | | | | | | | | | |
|---|---|---|---|---|---|---|---|---|---|---|
| | TM | MM | IC1 | IC2 | R1 | R2 | TG | UTG | NF | Mean |
| Soil pH | 6.94f | 7.31de | 7.98bc | 8.35ab | 8.50a | 7.79c | 7.91bc | 7.09f | 7.28e | 7.68 |
| EC (ds m$^{-1}$) | 0.057c | 0.333ab | 0.327ab | 0.510a | 0.273b | 0.390ab | 0.267b | 0.210bc | 0.230bc | 0.289 |
| CEC (cmol$_c$ kg$^{-1}$) | 18.8f | 37.8b | 34.9c | 32.2cd | 28.2d | 29.0d | 50.3a | 11.4e | 49.8a | 32.6 |
| Ex Ca (cmol$_c$ kg$^{-1}$) | 8.5f | 18.3b | 12.1e | 13.3d | 14.0cd | 14.8c | 24.8a | 4.5g | 25.5a | 15.1 |
| Ex Mg (cmol$_c$ kg$^{-1}$) | 2.7f | 8.8c | 6.3e | 6.5de | 7.1d | 6.8d | 14.7b | 1.3g | 15.1a | 7.6 |
| Ex Na (cmol$_c$ kg$^{-1}$) | 0.040f | 0.213d | 0.667a | 0.682a | 0.627b | 0.613b | 0.450c | 0.153e | 0.030f | 0.386 |
| Ex K (cmol$_c$ kg$^{-1}$) | 0.503d | 0.797b | 0.605c | 0.597c | 0.623c | 0.627c | 1.43a | 0.204e | 1.39a | 0.753 |
| SBFC (cmol$_c$ kg$^{-1}$) | 11.7e | 28.1b | 19.7d | 21.1cd | 22.4c | 22.9c | 41.4a | 6.2f | 42.0a | 23.9 |
| ESP | 0.12f | 0.45 | 1.40a | 1.26b | 1.21b | 0.97c | 0.84d | 0.28e | 0.06f | 0.73 |
| BSP | 62.5d | 74.3b | 68.2cd | 69.4b | 69.6b | 67.4d | 82.3a | 54.4e | 84.3a | 73.3 |
| SAR | 0.011f | 0.049 | 0.153a | 0.140ab | 0.139b | 0.117c | 0.098d | 0.034e | 0.007f | 0.083 |
| Pav (mg kg$^{-1}$) | 2.0gh | 13.8c | 2.8e | 3.1de | 3.5d | 4.0d | 19.4b | 1.4h | 23.9a | 8.2 |
| OC (%) | 0.643f | 1.540c | 1.250d | 1.220d | 1.110e | 1.113e | 3.520b | 0.413g | 4.980a | 1.754 |
| TN (%) | 0.067f | 0.135c | 0.119d | 0.116d | 0.108e | 0.109e | 0.257b | 0.034g | 0.385a | 0.148 |
| OC:TN | 9.597g | 11.407d | 10.50e | 10.51e | 10.28f | 10.21f | 13.696a | 12.147c | 12.935b | 11.87 |

Means followed by different letters in the same rows are significantly different at probability level (P) = 0·05.

pH, hydrogen ion concentration; EC, Electrical conductivity; CEC, cation exchange capacity; Ex Ca, Ex Mg, Ex Na, Ex K, exchangeable calcium, magnesium, sodium, potassium, respectively; SBFC sum of base forming cations; ESP, Exchangeable Na percentage; BSP, Base saturation percentage; SAR, sodium absorption ratio; Pav, available phosphorus; OC, Organic carbon; TN, total nitrogen; OC;TN, ratio of OC to TN.

TM, Teff (*Eragrostis tef* (Zucc) Trot*)* mono-cropping; MM, Maize (*Zea mays L.)* mono-cropping; IC1; Cauliflower (*Brassica oleracea var. botrytis*) with maize intercropping; IC2, Red beet (*Beta Vulgaris*) with maize intercropping; R1, Cauliflower – teff - maize rotation; R2, Onion (*Allium cepa* L.) - maize - onion rotation; TG, Treated gully; UTG, Untreated gully; NF, Natural forest.

assess the salinity status of sub-surface soil layers of the irrigated fields in order to understand the extent of salt leached towards the lower soil horizons.

**Effects on cation exchange capacity and base forming cations.** The highest CEC (50.3 cmol$_c$ kg$^{-1}$) and Ex K (1.43 cmol$_c$ kg$^{-1}$) were found from TG followed by NF (CEC = 49.8 cmol$_c$ kg$^{-1}$, Ex K =1.39 cmol$_c$ kg$^{-1}$) whereas the lowest CEC (11.4 cmol$_c$ kg$^{-1}$) and Ex K (0.204 cmol$_c$ kg$^{-1}$) were recorded from UTG. The significantly higher Ex Ca (25.5 cmol$_c$ kg$^{-1}$) and Ex Mg (15.1 cmol$_c$ kg$^{-1}$) were recorded from NF whereas the lowest Ex Ca (4.5 cmol$_c$ kg$^{-1}$) and Ex Mg (1.3 cmol$_c$ kg$^{-1}$) were observed from UTG (Table 4). The CEC, Ex Ca, Ex Mg, and Ex K recorded from MM were significantly higher than that of TM, IC1, IC2, R1, R2, and UTG. However, there were non-significant differences in CEC, exchangeable Ca, Mg, and K between the intercropping systems (IC1 and IC2), and also between crop rotation practices (R1 and R2) in the study sub-catchment. This indicates that crops used for intercropping and rotations have similar effect on CEC and soil exchangeable bases. However, CEC and those exchangeable bases recorded from irrigated fields with crop rotations were found to be higher than that of intercropping systems and rain-fed TM because crop rotation coupled with long-term soil management practices can positively affect soil OC and thereby CEC. Meaning soils with low OC can have a low CEC value and vice versa. Similarly, values of these soil properties were significantly higher in irrigated intercropping than rain-fed TM. Generally, CEC and exchangeable bases observed from NF and TG were categorized as very high; that of MM, IC1, IC2, R1 and R2 as high; from TM as medium; and UTG as a low rate as compared to the rates reported for African soils by Landon [81].

The highest Ex Na was found from IC2 (0.682 cmol$_c$ kg$^{-1}$) followed by IC1 (0.667 cmol$_c$ kg$^{-1}$); while the lowest Ex Na was recorded from NF (0.030 cmol$_c$ kg$^{-1}$) and TM (0.040 cmol$_c$ kg$^{-1}$). The Ex Na recorded from TG was significantly higher than that of NF, TM, MM and UTG (Table 4), in which this could be attributed to the effects of long-term irrigation water drained to TG as it is located in the depression site within the irrigation fields. There are white crust indicators on the irrigation fields that indicate the long-term application of irrigation water leads to the accumulation of salt on the soil surface just after the soil moisture used by the plant or lost by evaporation. Thus, drained water to TG sites can increase Ex Na as compared to the other CLUS. Such accumulations of salts can affect gradually the soil structure through time.

According to Landon [81], Ex Na observed from IC1, IC2, R1, R2 and TG were rated as medium; MM and UTG as low; and NF and TM as very low. However, the Ex Na from the irrigation fields was found to be near to the cut-off point for medium rate (0.7 cmol$_c$ kg$^{-1}$) which is regarded as potentially sodic, indicating that necessary soil and crop management practices should be taken to reduce or maintain Ex Na of the soil at the existing level. In addition, the highest SBFC and BSP were found from NF and TG whereas the lowest was from UTG followed by TM (Table 4). According to the report by Landon [81] the BSP from NF and TG was rated as very high and that of UTG was rated as medium. According to the same author, the BSP of the remaining CLUS were categorized in the high rate. The highest ESP and SAR were recorded from IC1 followed by R2 and the lowest was from NF followed by TM (Table 4). However, all the CLUS showed ESP < 2, which is classified as low or non-sodic soils as reported the rate for African soils by Landon [81]. Since the SAR is < 12 which is the cut-off point [58], the soil of the CLUS is categorized as non sodicity. This indicates that sodicity could not affect soil physical properties such as soil aggregation, infiltration of the irrigated cropping and other land use systems even though there is a trend of sodicty increment on the irrigation fields.

## Effects on soil nutrients

The highest and statistically significant Pav was recorded from NF (23.9 mg kg$^{-1}$) followed by TG (19.4 mg kg$^{-1}$). However, the lowest Pav was found from UTG (1.4 mg kg$^{-1}$) followed by TM (2.0 mg kg$^{-1}$). The Pav contents were non-significantly differed among the intercropping and crop rotation practices under irrigation system. But Pav from MM was found to be significantly higher than the other cropping systems (Table 4). Soil Pav of NF, TG, and MM were rated as very high, high and medium, respectively, and the rest CLUS rated as very low in Pav as compared to the rates reported by Landon [81]. Soil and crop management practices that improve soil nutrients should be given due attention to CLUS with very low soil Pav. Generally, this study result of Pav is consistent with previous reports elsewhere (e.g. Lemenin and Hanna [41]; Solomon et al. [84]; Nweke and Nnabude [85]; Flynn [86]) who stated that soil Pav variability has related to land-use type and soil management practices. For example, losses of Pav is higher in continuously cropped land as compared to forest land and other well managed land-use systems due to its poor fixation, removed with crop harvest, poor residual management and erosion processes [85, 86].

The highest and statistically significant soil organic carbon (OC) was found from NF (4.98%) followed by TG (3.120%) while the lowest OC was from UTG (0.413%) followed by TM (0.643%). The optimal OC, i.e., between 3% < OC < 5% as proposed by Craul [87], indicates low risk of soil structural degradation that is consistent with the values of NF and TG. The soil OC recorded from MM (1.45%) was significantly higher than that of intercropping (mean OC 1.31%) and crop rotation (mean OC 1.11%). The reason could be due to continuous

application of manure or compost to MM than in the other arable fields and its residual effects on the sol system. The soil OC was higher in the intercropping fields than in the crop rotation which could be associated with the effect of legume crop (*Guizotia abyssinica* L.) intercropped with tef during the rain-fed crop season. Long-term studies on the benefit of manures, inter-cropping, and crop rotation have consistently reported for maintaining and increasing OC inputs into the soil and thereby impacts on other soil properties [4, 14, 28]. However, continu-ous cropping and improper intercropped and crop rotated fields can be resulted in a decline in OC. The magnitude of soil OC reduction is also affected by tillage, management practices, and climate and soil conditions [14, 88].

According to Kay and Angers [89], irrespective of soil type and climatic condition if SOC contents are below 1% (e.g., TM, UTG), it may not be possible to obtain potential yields. Because SOC can impact on other physical, chemical and biological soil properties. In this study, SOC of TM and UTG showed very low; NF followed by TG showed very high and the SOC of the remaining CLUS (Table 4) are within the low rate as reported for African soils by Landon [81]. In agreement to the present finding other researchers elsewhere have reported that continuous cultivation depleted SOC and reduced soil quality compared to native vegeta-tion (NF), regardless of the cropping system practiced (e.g., Reeves [88]; Bowman et al. [90]; Bremer et al. [91]).

The highest soil total nitrogen (TN) was found due to NF (0.541%) followed by TG (0.257%); whereas the lowest was recorded from UTG (0.030) followed by TM (0.067%). The soil TN from MM was significantly higher than that of IC1, IC2, R1 and R2 (Table 4), which could be attributed to the higher amount of manure applied to MM fields continuously during each cropping seasons. TN from NF and TG rated as very high and high, respectively. The TN from the other CLUS rated as low except MM which is rated as medium; and UTG as very low based on the rate described by Landon [81]. The present finding on OC and TN is consistent with other reports which have reported that SOC and TN content are not only affected by cli-mate and terrain, but also by land-use and soil management practices. For example, agricul-tural intensification and repeated cultivation have resulted in a serious decrease in SOC and TN as compared to natural vegetation such as NF. The fact is that cultivation enhances decom-position of soil OC, and physical removal of biomass as straw and grain harvest reduces its availability (e.g., Zhang et al. [61]; Wu et al. [63]; Nweke and Nnabude [85]; Johnson and Cur-tis [92]).

## Effects on OC to TN ratio

The C:N ratios of the soils in the different CLUS range from 9.6 to 13.7. The highest C:N ratio was recorded from TG (13.7) followed by NF (12.9) and UTG (12.2) (Table 4), which could be associated with low oxidation (decomposition) rate of organic sources as compared to the inputs available in the study sub-catchment. Meaning, there were no soil and agronomic prac-tices that enhance decomposition of organic sources in these selected land-use systems. In addition, the soil in TG was water saturated almost for more than 6 months of the year, in which this could be slow down the decomposition of organic matter by limiting soil microbial activity [61, 63]. Under this condition of TG, slowing the decomposition of organic matter leads to have poor soil structure and high bulk density and low availability of soil nutrients. However, in the remaining four months the area under TG is getting free from water satura-tion in which this creates a favourable condition for organic matter decomposition.

Similarly, the long-term effects of irrigation practices can reduce microbial activity as satu-rated soil water creates unfavourable conditions such as water logging and low soil tempera-ture and thereby this reduces organic matter decomposition. This could be the reason for

having C:N ratio to be slightly higher than 10 in the irrigation fields such as IC1, IC2, R1 and R2. The C:N ratio of MM (11.4) was found to be higher than that of the fields under irrigation cropping systems. The reason could be associated with the application of higher manure on MM field continuously during the past several cropping seasons. The lowest C:N ratio was reported from TM (9.6), indicating that there is a higher organic matter mineralization. The extreme C:N ratio of 9.6 (TM) may also reflect the annual N fertilization that might be greater than the crop needs. The other extreme value of C:N ratio of 13.7 (TG) may reflect no outside fertilization and inputs are from the natural biomass such as the grasses growing on the land. TM was the only CLUS which showed C:N ratio below 10 (Table 4), indicating a balance of soil carbon and nitrogen source inputs vs. decomposition as outputs ($CO_2$ and mineral nitrogen to crops or leaching) as mediated by soil microorganisms [61, 93].

An optimum temperature and moisture conditions could be enhanced microbial activities to decompose organic sources in the fields such as the TM [63]. However, the selected fields did receive little moisture from rainfall for about 8 months. The conventional tillage practice (cultivation) used in TM could also enhances organic sources to be decompose quickly [93, 94]. In cropping systems such as TM, tillage practice coupled with insufficient inputs of organic sources and removals of crop residues in the absence of crop rotation resulted in a lower C:N [62, 95]. From agricultural production point of view literature showed that cropping and land-use systems with C:N ratio < 10 is rated as good, 10.1-14 as medium and > 14 as poor soil systems [81]. When C:N ratios are near normal, then minor differences in C:N ratio can have very little effect on soil productivity in which the implication is that soil productivity is influenced by other productivity factors such as water availability in the soil system. However, such values are contrasted to the present pressures to reduce carbon emission to the atmosphere and sequestered carbon through maintaining higher soil C:N ratio [62, 96, 97]. Hence, the CLUS such as TG and NF showed better or conducive conditions for carbon sequestration as organic matter decomposition can slowdown by environmental conditions that are limiting microbial activity as compared to the cropping systems (e.g., TM, MM) which accelerate decomposition of organic sources. The C:N ratio determined from TG and NF is lower than that of apple orchard (C:N ratio of 15.4) reported from the northern China [62, 97]. However, the existing reports have not quantified the contribution of nitrogen deposition from the atmosphere to soil TN as this is the other factor that may potentially affect the results of soil C:N ratio of the different CLUS.

## Effect on soil organic carbon (OC) and total nitrogen (TN) stocks

There is variability in the soil mass at fixed depth of 0-20 cm and also for the genetic horizon among the different CLUS. As a result, the soil OC and TN stocks calculated using the equivalent soil mass was used to standardized the stocks comparison among the CLUS as it is presented in Table 5. In the study sub-catchment, soil OC and TN stocks varied significantly among the majority of the CLUS. The highest stock of OC (175.3 Mg C $ha^{-1}$) and TN (13.6 Mg C $ha^{-1}$) were reported from NF. This was followed by TG of OC and TN stocks of 123.9 and 9.05 Mg $ha^{-1}$ respectively. The soil OC stock from NF was found to be higher than that of reported for tropical forests (122 Mg C $ha^{-1}$) by Prentice et al. [98] in which such differences could be attributed to variability in N-fixing trees, soil factor and climatic conditions. The lowest stocks of soil OC (14.5 Mg C $ha^{-1}$) and TN (1.20 Mg C $ha^{-1}$) were found from UTG (Table 5). The soil OC (123.9 Mg $ha^{-1}$) and TN stocks (9.05 Mg $ha^{-1}$) of TG were significantly higher than the other CLUS, except that of NF. In line with the present results, Lal [25] has reported that soil organic matter (OM) can be greatly enhanced when degraded soils and ecosystems are restored, or converted to a restorative land-use or replanted to perennial

**Table 5. Mean soil masses of the generic horizon, masses in an equivalent mass of soil, SOC and TN stocks and reduction in stocks due to the different cropping and land-use systems as compare to the reference CLUS in the Dura sub-catchment, northern Ethiopia.**

| Cropping and land-use system | Soil mass (Mg ha$^{-1}$) at fixed depth of 0.20 m | Thickness of genetic horizon (m) | Soil mass of genetic horizon (Mg ha$^{-1}$) | Thickness to attain ESM (m) | Equivalent soil mass (ESM) (Mg ha$^{-1}$) | SOC stock (Mg C ha$^{-1}$) | SOC stock reduction (%) | TN stock (Mg N ha$^{-1}$) | TN stock reduction (%) |
|---|---|---|---|---|---|---|---|---|---|
| TM | 3200 | 0.150 | 2400 | 0.220 | 3520 | 22.6e | 87.1a | 2.36e | 82.7b |
| MM | 2800 | 0.171 | 2394 | 0.252 | 3520 | 54.2c | 69.1c | 4.75c | 65.1e |
| IC1 | 2940 | 0.160 | 2352 | 0.239 | 3520 | 44.0d | 74.9b | 4.19d | 69.2d |
| IC2 | 3200 | 0.150 | 2400 | 0.220 | 3520 | 42.9d | 75.5b | 4.08d | 70.0cd |
| R1 | 2860 | 0.168 | 2402 | 0.246 | 3520 | 38.9d | 77.8b | 3.80d | 72.0c |
| R2 | 2880 | 0.167 | 2405 | 0.244 | 3520 | 39.2d | 77.7b | 3.84d | 71.8c |
| TG | 2640 | 0.182 | 2403 | 0.266 | 3520 | 123.9b | 29.3d | 9.05b | 33.5f |
| UTG | 3520[a] | 0.136 | 2394 | 0.200 | 3520 | 14.5f | 91.7a | 1.20f | 91.2a |
| NF | 2400 | 0.200 | 2400 | 0.293 | 3520 | 175.3a | - | 13.6a | - |
| Mean | | | | | | 61.8 | 64.8 | 5.20 | 61.8 |

TM, Teff *(Eragrostis tef* (Zucc) Trot*))* mono-cropping; MM, Maize *(Zea mays L.)* mono-cropping; IC1; Cauliflower (Brassica oleracea var. botrytis) with maize intercropping; IC2, Red beet *(Beta Vulgaris)* with maize intercropping; R1, Cauliflower – teff - maize rotation; R2, Onion *(Allium cepa* L.)- maize - onion rotation; TG, Treated gully; UTG, Untreated gully; NF, Natural forest land system.

[a]The UTG land use and cropping system was used as the equivalent soil mass (heaviest soil) for the fixed depth of 0.20 m in which the other CLUS soil thickness that required to attain this mass was calculated.

vegetation (e.g., TG); and depleted OM in agricultural soils that use conventional tillage (e.g., TM). This could be the reason for OC to be used as an important indicator of both soil productivity and climate change mitigation interventions [25, 38].

The soil OC and TN stocks estimated from MM were showed significantly higher than all the other CLUS, except that of NF and TG. This could be associated with the continuous seasonal application of organic inputs on the MM fields. The soil OC and TN stocked in the intercropped fields were higher than that of rotation and TM in which this might be associated with the relatively effectiveness of the intercropping system for improving soil OC sequestration. The present results of OC and TN stocks variability across the CLUS is agreed with the previous reports of Bird et al. [99] and Saiz et al. [100] who have reported variability in soil nutrient stocks could be attributed to differences in soil and crop management practices across the different CLUS. For example, there are practices in NF and TG (Table 1) that reduce soil OM mineralization and erosion and also increase OM inputs to the soil in which this could improve soil OC and TN stocks.

As compared to the reference land-use system (NF), soil OC stock due to UTG and TM was reduced by 91.7 and 87.1%, respectively. Similarly, TN stock was reduced due to UTG and TM by 91.2 and 82.7%, respectively. The mean reduction in OC stocks by crop rotation, intercropping, MM, and TG as compared to the NF were found to be 77.8, 75.2, 69.1, and 29.3%, respectively. The mean TN stock reduced from fields with crop rotation, intercropping, MM, and TG as compared to NF was reported to be 71.9, 69.6, 65.1, and 33.5%, respectively. This study results indicate that the SOC and TN stocks are drastically reduced in most of the CLUS, with the highest reduction was observed from UTG followed by TM. The variation in nutrient stocks under such CLUS should be associated with the differences in soil and crop management practices; e.g., no organic fertilizer was applied on UTG and TM; besides to the nature of the CLUS to resist against erosion. Such losses of OC and TN from the soil system could increase the amount of carbon and nitrogen gasses in the atmosphere at global-scale [38, 101].

The OC stock reduction is slightly higher than the TN stock across all the CLUS except TG, indicating that the need for understanding the reason using further investigation.

Generally, the soil OC stock reported in this study is found to be higher than the previous reports from Africa (e.g., Amudson [101]) who has reported that cultivation reduces the original OC content by up-to 30%. In the study catchment, such variability could be attributed to the differences in the duration and type of the management practices implemented in the different CLUS. For example, the arable lands such as TM have been cultivated for more than 100 years using organic fertilizer in the study sub-catchment. In addition, increasing soil OC stock is a major challenge in the dry areas where vegetation growth is hampered by climatically harsh conditions such as low availability of water and nutrients, removal of fodder and fuelwood [38].

## Determinants of soil properties variability using PCA

The bivariate correlation analysis among many soil properties determined from the CLUS correlated (r) at r > 0.70 which qualitatively described as moderate to extremely strong correlation (data not shown). Such values of r indicate that there are multicollinearity effects among the soil properties. The effect of multicollinearity was handled using the partial correlation and principal component analysis (PCA) that grouped soil properties into five principal components (PCs) (Table 6). The eigenvalue of PC1, PC2, PC3, PC4 and PC5 are 8.50, 6.48, 2.24, 1.61 and 1.13, respectively. The five PCs that received eigenvalues >1 explain largely the variability of the soil properties among the CLUS [66]. The variances explained by PC1, PC2, PC3, PC4 and PC5 are 30.65, 24.32, 17.29, 7.90 and 6.63%, respectively, in which this is accounted for a total of variance explanation by 87%. Such variance values are in line with the report of Brejda et al. [66] who stated that PCs that receive at least 5% variance explains the best variability of a factor component. The soil properties included in the first five PCs explain for 86.8% of the soil quality variability among the CLUS. The communalities of the extracted five PCs explained by each soil property ranges from 74-96% (Table 6). A high communality variable shows that a high portion of variance was explained the variable and therefore, it gets a higher preference to a low communality [102].

The highly loading variables in PC1 were CEC and clay content (Table 6). Since the correlation coefficient between CEC and clay was 0.86 which is higher than the cut-off point (0.60), communality was considered to select the parameter to be retained in PC1. As a result, CEC was retained in PC1 because the loading (0.87) and communality (0.95) of CEC were higher than that of clay. The first PC is thus termed as '*cation exchange capacity*, *CEC factor*'. Similarly, the highly loading variables under PC2 were SOC, TN and soil structural stability index (SSSI). However, the partial correlation analysis results indicated strong correlations (r > 0.80) among these variables. Considering the higher partial correlation coefficient, loading value and communality, SOC was retained in PC2. The content of SOC influences directly the value of the other highly loading variables (TN and SSSI) [83, 103, 104]. The implication of such reports is that the contribution of TN and SSSI to PC2 is well explained using SOC. As a result, PC2 is termed as the '*organic matter factor*'.

The variability in PC1 and PC2 is not necessary to be explained due to the soil inherent properties. Rather it is mainly related to the effects of the long-term crop and soil management practices practiced on the different CLUS in which such practices can influence on soil erosion detachment and transportation processes. Such processes can practically change drastically the soil texture and other basic soil properties as fine soil particles are easily transported by leaving course particles in the source site. Soil nutrients which are attached with the soil particles are transported by erosion processes and so resulted in variability soil properties among the CLUS [70].

**Table 6. Results of principal component analysis of soil properties responses to the different cropping and land-use systems in the Dura sub-catchment, northern Ethiopia.**

| Principal component, PC | PC1 | PC2 | PC3 | PC4 | PC5 | |
|---|---|---|---|---|---|---|
| **Eigenvalue** | **8.50** | **6.48** | **2.24** | **1.61** | **1.13** | |
| Variance (%) | 30.65 | 24.32 | 17.29 | 7.90 | 6.63 | |
| Cumulative variance (%) | 30.65 | 54.97 | 72.26 | 80.16 | 86.79 | |
| **Variables** | **Eigenvectors** | | | | | **Communalities** |
| Cation exchangeable capacity | **0.87** | 0.01 | 0.27 | 0.19 | -0.20 | 0.96 |
| Clay | **0.75** | 0.07 | 0.17 | -0.27 | -0.04 | 0.92 |
| Silt | 0.54 | 0.23 | -0.01 | 0.00 | -0.34 | 0.85 |
| Sand | 0.40 | 0.14 | -0.20 | 0.33 | 0.19 | 0.78 |
| Soil organic carbon | 0.51 | **0.83** | -0.13 | -0.02 | 0.03 | 0.93 |
| Soil total nitrogen | 0.48 | **0.81** | -0.14 | 0.001 | **0.86** | 0.91 |
| Soil structural stability index | 0.31 | **0.79** | 0.52 | 0.09 | -0.03 | 0.86 |
| Dry bulk density | -0.16 | -0.51 | **-0.78** | -0.13 | -0.10 | 0.89 |
| Total porosity | 0.16 | 0.50 | **0.78** | 0.13 | 0.10 | 0.89 |
| A-horizon depth | 0.13 | 0.37 | **0.76** | 0.06 | 0.07 | 0.87 |
| Exchangeable Sodium | 0.23 | -0.28 | -0.14 | **-0.74** | -0.02 | 0.89 |
| Exchangeable Sodium Percentage | 0.15 | -0.21 | -0.12 | **-0.72** | -0.03 | 0.87 |
| Sodium adsorption ratio | -0.17 | -0.23 | -0.15 | **-0.71** | -0.06 | 0.85 |
| Available phosphorus | 0.47 | 0.25 | 0.15 | 0.18 | **0.81** | 0.90 |
| Soil pH | 0.29 | 0.22 | 0.32 | 0.21 | 0.52 | 0.75 |
| Exchangeable Calcium | 0.58 | 0.20 | 0.12 | -0.31 | 0.03 | 0.87 |
| Exchangeable Magnesium | 0.53 | 0.18 | 0.16 | -0.26 | 0.37 | 0.85 |
| Exchangeable potassium | 0.55 | 0.15 | 0.24 | 0.17 | 0.45 | 0.87 |
| Electrical conductivity | 0.52 | 0.02 | 0.23 | 0.56 | 0.34 | 0.74 |
| Sum of base forming cations | 0.42 | 0.07 | 0.19 | 0.20 | 0.17 | 0.85 |
| Base saturation percentage | 0.20 | 0.16 | -0.11 | -0.07 | 0.20 | 0.90 |
| Ratio of OC to TN (OC: TN) | 0.50 | 0.34 | 0.22 | -0.23 | 0.26 | 0.82 |

Underlined boldface eigenvector values corresponded to those highly weighted variables and retained in the PC.

The highly loaded variables in PC3 included dry soil bulk density (DBD), porosity and A-horizon depth (A_h). The partial correlation analysis between DBD and porosity showed at r = 0.85. Since the loading value and communality of DBD was slightly higher than that of the soil porosity, DBD was retained in PC3. A-horizon depth was also retained in PC3 as the correlation coefficient with the other high loading variables showed less than the cut-off point (< 0.60). Thus, PC3 is termed as '*soil physical property factor*'. The highly loaded variable in PC4 included Ex Na, ESP and SAR. The partial correlation values among these variables showed strong correlation (r > 0.88). Considering the higher correlation coefficient, loading weight and communality values, Ex Na was retained in PC4 and the rest variables were excluded from PC4. As a result, PC4 was termed as the '*sodicity factor*'.

Likewise, the highly loaded variables in PC5 are TN and Pav (Table 6). Since the correlation between TN and Pav is below 0.6, both variables were retained in PC5. The loading coefficient of TN in PC5 (0.86) is higher than in PC2 (0.81) in which this is another reason to retain TN in PC5. Nitrogen and phosphorus are also commonly reported as the most crop limiting soil nutrients in developing countries. So, considering these parameters in PC5 is important while assessing soil degradation using soil properties as an indicator among the CLUS. Thus, PC5 is termed as the '*soil macro-nutrient factor*'.

The seven PCA chosen final soil properties that better explain (determinant) of soil quality variability for 87% among the CLUS were CEC, SOC, DBD, A_h, Ex Na, TN, and Pav. Future assessment and monitoring of the effects of similar CLUS on soil quality is suggested to depend on these seven soil properties as these are sensitive to disturbances of the different land-use and management practices. This can reduce wastage of resources (e.g., cost, time, labor) while analyzing the entire datasets and also gives rapid response for effective assessment and decision-making while monitoring changes related to the different CLUS.

Generally, among the seven PCA chosen soil properties of this study, five (CEC, SOC, TN, phosphorus, DBD) parameters were similarly selected by Tesfahunegn et al. [83] who have reported from exclusively rain-fed land-use and soil management practices in the northern Ethiopia. The choice of SOC in the PC factor as determinant soil property for soil quality variability across the CLUS is consistent with other reports (e.g., Larson Pierce [105]; Shukla et al. [106]) who have reported that soil organic matter (SOM) is among the most powerful soil properties that influence many soil conditions. For example, Larson and Pierce [105] have reported that SOM improves the soil to accept, hold, and release nutrients, water and other soil chemical ingredients to plants, recharge surface water to groundwater; support root growth through soil structure stability, maintain soil biotic habitat; and resist soil degradation. The selection of DBD using the PCA could also confirm the basic principle of the soil to restrict water flow and plant root growth when DBD increases [107, 108]. However, climate change has reported to likely affect species future distributions of the different CLUS even in the global climate change scenarios [109, 110].

## Conclusions

This study revealed that soil properties and nutrient stocks are significantly varied across the different cropping and land-use systems (CLUS) in the study sub-catchment. Such variation found to be attributed mainly to the difference in the CLUS. The natural forest (NF) followed by the treated gully (TG) were attempted to maintain the sustainability of the soil properties as compared to the other CLUS. All the soil chemical properties except exchangeable sodium (Ex Na) showed the highest values in NF and TG. Similarly, NF followed by TG showed a lower dry bulk density than the other CLUS. Conversely, the soil properties and nutrient stocks determined from the untreated gully (UTG) followed by the tef mono-cropping (TM) showed their soil quality is seriously degraded. The mean soil nutrient stocks as OC and TN is reduced to the extent of 91.5% and 84.9% of the reference less disturbed soil in UTG and TM, respectively. The soil properties and the OC and TN stocks status determined from the maize mono-cropping (MM) is by far better than that of theTM, irrigated intercropping and crop rotation fields. The highest Ex Na was reported from the irrigation fields, particularly from the inter-cropping systems in which this indicates that there is a need to give serious attention on activities related to such cropping system so that to maintain the soil sodium status at least at its current level. However, the other soil properties and nutrient stocks (SOC and TN) showed a better improvement in intercropping than crop rotation fields. Generally, this study revealed that continuous irrigation lowers the values of the soil variables and soil nutrient stocks as compared to the practices in MM and NF, implying that such irrigation practices should be supported by suitable soil and cropping systems integrated with efficient water usage in order to achieve a sustainable and high agricultural production.

In this study, the soil properties values generally showed an early warning about the severity of soil physical and nutrient degradation in some of the CLUS such as UTG and TM followed by the irrigated fields. On the other hand, the CLUS having a better soil prosperities and nutrient stocks were reported in descending order as NF, TG and MM. It is thus suggested for the

implementation of appropriate intercropping and crop rotation systems (e.g., use of legume crops and trees) and best soil management practices (optimum rate of integrated organic and inorganic, erosion controlling) that improve and maintain the soil quality of the CLUS in the study sub-catchment conditions. For further monitoring of the effects of the CLUS on soil quality in the study sub-catchment conditions, attention should be given to the PCA selected variables (CEC, SOC, DBD, A_h, Ex Na, TN, and Pav) that explained for 87% of the soil quality variability is due to the CLUS. Such selections of few relevant soil parameters are helpful as a strategic approach to assess soil properties that leads to quick monitoring (rapid and inexpensive) and effective decision-making against the effects of each CLUS on soil properties and nutrient stocks in the study sub-catchment conditions.

## Supporting information

**S1 Data.**
(XLS)

## Acknowledgments

This research was financial supported by University of Aksum (Ethiopia) under the terms of grant no. AKU/IG/RCSD/1092/07. The author gratefully acknowledged the financial support provided by Aksum University to conduct this study. The author also highly appreciated the farmers and development agents who involved in the identification and characterization of the different cropping and land-use systems. The support provided during the soil sample collection by Mr. Kahsu Kidane (Development Agent) is highly appreciated. The assistance offered by the village administration and development agents during all the discussions and data collection processes are also highly acknowledged.

## Author Contributions

**Data curation:** Gebreyesus Brhane Tesfahunegn, Teklebirhan Arefaine Gebru.

**Formal analysis:** Gebreyesus Brhane Tesfahunegn.

**Funding acquisition:** Gebreyesus Brhane Tesfahunegn.

**Investigation:** Gebreyesus Brhane Tesfahunegn, Teklebirhan Arefaine Gebru.

**Methodology:** Gebreyesus Brhane Tesfahunegn.

**Project administration:** Gebreyesus Brhane Tesfahunegn.

**Resources:** Gebreyesus Brhane Tesfahunegn.

**Software:** Gebreyesus Brhane Tesfahunegn.

**Supervision:** Gebreyesus Brhane Tesfahunegn.

**Validation:** Gebreyesus Brhane Tesfahunegn.

**Visualization:** Gebreyesus Brhane Tesfahunegn.

**Writing – original draft:** Gebreyesus Brhane Tesfahunegn.

**Writing – review & editing:** Gebreyesus Brhane Tesfahunegn, Teklebirhan Arefaine Gebru.

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
