## [Decision Letter · Decision Letter 0]

1 Oct 2019

PONE-D-19-24034

Variation in Soil Properties under Long-Term Irrigated and Non-Irrigated Cropping and Other Land-Use Systems in Dura Catchment, Northern Ethiopia

PLOS ONE

Dear Dr. Tesfahunegn,

Thank you for submitting your manuscript to PLOS ONE. After careful consideration, we feel that it has merit but does not fully meet PLOS ONE’s publication criteria as it currently stands. Therefore, we invite you to submit a revised version of the manuscript that addresses the points raised during the review process.

Both reviewers agree that your work is interesting but they feel that other factors than irrigation, such as soil properties and agricultural practices, need to be considered for the interpretation of the results. Please take their comments in this direction very seriously when revising your manuscript.

We would appreciate receiving your revised manuscript by Nov 15 2019 11:59PM. To enhance the reproducibility of your results, we recommend that if applicable you deposit your laboratory protocols in protocols.io, where a protocol can be assigned its own identifier (DOI) such that it can be cited independently in the future. For instructions see: http://journals.plos.org/plosone/s/submission-guidelines#loc-laboratory-protocols

We look forward to receiving your revised manuscript.

Kind regards,

Remigio Paradelo Núñez

Academic Editor

PLOS ONE

Journal Requirements:

1. In your Methods section, please provide additional location information of the sampling sites, including geographic coordinates for the data set if available.

2. In your Methods section, please provide additional information regarding the permits you obtained for the work. Please ensure you have included the full name of the authority that approved the sampling sites access and, if no permits were required, a brief statement explaining why.

Additional Editor Comments (if provided):

Both reviewers have raised concerns regarding the fact that important information about the soils and the agricultural systems and practices have not been taken into account for the interpretation of your results, suggesting that irrigation might not be the most important factor explaining the differences in soil quality. I agree with them, so I would suggest to revise your work taking into account their comments in this direction.

Reviewers' comments:

Reviewer's Responses to Questions

**Comments to the Author**

1. Is the manuscript technically sound, and do the data support the conclusions?

Reviewer #1: Partly

Reviewer #2: Partly

2. Has the statistical analysis been performed appropriately and rigorously? 

Reviewer #1: Yes

Reviewer #2: No

3. Have the authors made all data underlying the findings in their manuscript fully available?

Reviewer #1: Yes

Reviewer #2: Yes

4. Is the manuscript presented in an intelligible fashion and written in standard English?

Reviewer #1: Yes

Reviewer #2: Yes

5. Review Comments to the Author

Reviewer #1: This manuscript reports an attempt to identify key factors affecting soil quality (productivity) in a region where little is known about the interaction of soil properties and soil quality in small farmer agricultural production systems. Specific Comments are:

1. Significant additional editing of English usage and grammar are needed throughout the manuscript.

2. Page 3, line 79: Change “have practiced” to “have been implemented”.

3. Page 5, line 116: Change “to deal” to ‘dealing”.

4. Page 5, line 123: Insert “have” between “that” and “generalized”.

5. Page 5, lines 129-132: “…cannot be said…”? English editing???

6. Page 5, lines 137-142: (Objectives) Are these properties to be related to specific land use/management practices? Is this information intended to be used only in the Dura sub-catchment or also used in other areas of Ethiopia? It might make this study more valuable in the authors could indicate if this concept is a “stand alone” effort for a small part of Ethiopia or whether it is part of a larger plan or concept for improving the agricultural lands in Ethiopia. Or, is it designed to specifically for education of small farmers. Placing this study into a larger context makes this work more valuable.

7. Page 6, line 170: What is an “irrigation command area”? Is it an irrigation project area? Is it an area where irrigated agriculture predominates? Is this an area where irrigation water is available but is used at the choice of the farmer? Is part of this study designed to evaluate the impacts of irrigation on soil quality outside of just making more water available for crops? An explanation of how this fits into the overall agricultural picture in the sub-catchment would provide a clearer context for the conduct of the study.

8. Page 12, lines 328-336: It is not clear from the description of the sites what the actual soil types and parent materials exist at each sampled site. Differenced is sand can be due to erosion, deposition of erosion sediments or original soil parent material. The high clay component for the R1 and R2 systems are probably due to being in a more level depositional or alluvial landscape and may represent the Chromic Vertisols. Because they are nearly level and higher in clay, they are most suitable for irrigated agriculture. Care must be taken in interpreting data from a study like this because the properties of the soils may not be due to land use but rather the land use may be due to soil properties. A danger of this type of study is that it may be inappropriately interpreted if it is generalized over a larger area. The nature of soils in an area usually dictates how they are used. Also, use of fertilizer doesn’t usually affect soil texture. Inherent soil properties and erosion might be responsible, however. The fertilizer might have some effect on soil structure but not on soil texture.

9. Page 12, lines 337-347: See the previous comment.

10. Page 15, lines 429-445: The SAR of all of these soils in <1which means that Na is not a problem in the soil of the area, The authors indicate that the higher Ex Na in the TG is due to irrigation water runoff into the TG. However, water runoff into TG may or may not be responsible for the higher Ex Na. If it is, then the authors should show a link to the Na in the irrigation water (water quality?). It could also be due to the properties of the parent material.

11. Page 17, lines 507-509: Irrigation may also improve microbial activity because of more soil water conditions. This is difficult to interpret from C:N ratios. The C:N ratio of these soils are in the range of 10-12 which is quite normal for most soils. The TM (lowest) and TG (highest) may be the only systems that reflect the management systems but needs and examination of C and N inputs and erosion in these systems.

12. Page 19, lines 563-566: This is difficult to interpret without knowing present and past erosion history.

13. Page 19, line 580: Change “soil prosperities” to “soil properties”.

14. Table 1: The rates of fertilizer need to be discussed more clearly either in the table or in the text. Potentially, information about whether DAP and urea were supplied annually, for each crop if double cropped or only occasionally within a cropping system is important in order to interpret the N and C status of the soils. “SWC practices were used at field borders” - What does this mean?

15. Table 6: Generally in summary, the clay/CEC and SOM components explain most of the differences between the systems. Right? The first two PCA’s relate to the inherent soil properties and the last three relate to management. This again illustrates the need for knowledge of distinct soil types because these are mainly responsible for soil properties across management systems.

The authors need to revise their manuscript with consideration of the above comments before it is accepted for publication.

Reviewer #2: General comments

This work describes in detail the results of a survey conducted to identify contrasting agrosystems and other land uses in a small catchment in Northern Ethiopia, including irrigated and rainfed systems, a reference forest soil and eroded areas with and without reclamation measures.

In this sense, the study provides detailed information on the nine systems selected, both on terms of management and about soil physical and chemical properties of the studied upper layer (0-20 cm). It adds this valuable information to the existing knowledge on land uses in the region.

From these data, authors describe the major characteristics of the soils under these managements, and use PCA to depict those properties explaining better the observed variability, with the aim to identify useful soil quality indicators for further studies. This approach is interesting as it helps also summarizing the information obtained about the 28 soil properties studied.

From this, the study seems a good contribution to the knowledge of soil condition in the study area, and to the relationship between land uses and soil properties. From this perspective, it is therefore worth publication, although, to my view, there are some issues that would need to be addressed before acceptation:

As a major concern, the results of the different properties analyzed are studied as if the only factor possibly inducing differences was the type of soil management. This was however probably not the case, as significant differences were observed in relatively unchangeable soil properties such as texture. Accepting that some soil management conditions leading to intense erosion can result in textural changes at the topsoil (lines 330-332), the observed differences in clay content (from 73.7% in R2 to 26.3 under TM, Table 3) are very likely due to soil heterogeneity. Although the authors declare that all the plots selected were on the same type of soil (Chromic Vertisol), this is information is not clear enough to warrant similar soil characteristics in all plots. This seems therefore a study based in pseudo-replication, and this should be accounted for in the description of results and the discussion. As soil texture is determinant for many of the studied soil properties such as CEC, this heterogeneity imposes limitations to the comparisons among sites. This is acknowledged by the authors (line327) but not really considered in the discussion.

In relation to the objective, although irrigation as a soil management system that can induce changes in soil condition is commented with some detail in the introduction, and included in the title, the study does not allow to draw clear conclusions on the effect of irrigation per se, as it compares irrigated and non-irrigated systems which differ in many other factors. For instance, the non irrigated systems (TM and MM) are monocultures, and the irrigated systems include intercropping and rotations, but not monoculture. Also MM and TM differ in the use of organic amendments in MM but not TM. Therefore, it seems more logical to compare the agrosystems between them (as such), but not trying to derive conclusions on the effect of irrigation as an isolated factor. My suggestion would be to remove from the title the idea that the study compares irrigated vs non-irrigated systems, and focus on the fact that it makes a comprehensive revision of the most common agrosystems and land condition (forest, gullies) found in the region.

Some of the properties studied, and in particular aggregate stability (but also SAR and porosity for instance) were not directly analyzed, but derived from others in the dataset. This needs to be handled with care when studying correlation in the PCA, as of course variables calculated from other variables would correlate with them. In particular, the formula used to calculate aggregate stability needs to be revised (line 248): as it is written, under equal SOC concentrations, this expression would result in a lower value (less stability) for a soil with more clay and silt (less sandy) than to a sandy soil with less clay + silt. This is counterintuitive and probably the cause of some unexpected results in this parameter found in Table 3.

The calculation of SOC stocks for comparing (supposedly) the same soil under different management practices needs an approach based in comparing equivalent masses (i.e., the amount of SOC in the same mass of soil for all managements considered). Although the reference used to describe the methodology for SOC storage does in fact describe this point (Ellert & Bettany, 1995), the authors did not use this approach. As differences were observed in bulk density, using the same formula provided in line 274 for two soils with the same SOC concentration but different bulk density would result in a greater stock in the one with higher bulk density. To avoid this bias, the approach of an equivalent mass comparison requires some easy calculations to correct the depth (T) considered in each case.

As a summary, my recommendation is that the authors consider these aspects and run the comparative study with the new perspective and data, including PCA.

I hope that these comments, which are meant to be constructive, are of use for a fruitful revision of the manuscript.

Some other minor comments are:

Introduction:

L.75-76: Irrigation can indeed increase SOC by increasing primary productivity, but contrasting results have been reported in this sense (see for instance recently daGama et al., 2019. Agronomy 2019, 9, 132; doi:10.3390/agronomy9030132).

L.103-109: This paragraph would be better in the paragraph in lines 61-62, as it contains very similar information.

L.82. Soil organic C is a nutrient? For what organisms? For plants?

L.137-142. See previous comment on the problems to compare irrigation as a factor in this study in relation to objective one. Suggest to replace by “under different cropping and other land-use systems”.

Mat-met

L.162. Same comment on irrigated and non-irrigated. Why stressing this and not for instance adjacent mono and poli-culture systems?

L.209-211. The reader has to rely about the homogeneity of the different areas based on this statement. However, results (Table 3) indicate that soils were different at least in one genetic characteristic such as topsoil texture... This should be considered in the discussion.

L.226 no need to indicate irrigated and non-irrigated, as undisturbed soil samples were also collected from the native forest and the unreclaimed gully for instance...

L.247-254. See previous comment on this index for soil structural stability. If this parameter is not to be measured directly on soil samples, to my view, better formulas are available in the literature to relate texture and organic matter to aggregate stability, that consider the role of BOTH organic matter and soil clays in structure stabilization.

L.272-280. See comments above on the use of equivalent masses to compare SOC (and TN) stocks.

L.285-289. Without commenting on the adequateness of ANOVA for this treatment, management systems are used here as one factor, without separating “irrigation” as an independent factor. This supports my previous observations that irrigation should not be commented in this work as a factor of study. It just makes part of some of the agrosystems studied, and it can be used to explain the results, just as organic fertilization, afforestation, tillage and the other parameters included in the management routines of the agrosystems described in Table 1.

Resuls & Discussion

L.322-327. See comments above on the difference sin texture. It seems unlikely (and cannot be proved with this study) that soil management is the responsible of all the differences observed in clay content among the nine studied sites.

L.390-391. Why should irrigation increase the soil pH? Is there anything in irrigation water quality that suggests so? Why it would only affect R1 and not the other irrigated treatments?

L.421-422. What is the relationship between between soil management and CEC?

L.429-445. Some discussion is done here on NA and SAR. I agree that the values reported are not within the range of sodicity, but could the different values observed between treatments affect for instance soil aggregation and therefore infiltration, clogging, etc?

L.505. So we discover here that the “non-irrigated” TM treatment is water-saturated more than 8 months per year. How can this have changed soil properties other than SOC?

L.534-576. See comments above on the comparisons on SOC that require equivalent mass corrections.

L.580 Soil prosperities? (guess properties...)

L. 596-623. See previous comment on covariance and correlation of variables derived mathematically from each other.

Conclusions

Can be revised after addressing previous comments.

6. PLOS authors have the option to publish the peer review history of their article (what does this mean?). If published, this will include your full peer review and any attached files.

Reviewer #1: No

Reviewer #2: No

---

## [Author Response · Author response to Decision Letter 0]

21 Oct 2019

Journal: PLOS ONE

PONE-D-19-24034

Variation in Soil Properties under Long-Term Irrigated and Non-Irrigated Cropping and Other Land-Use Systems in Dura Catchment, Northern Ethiopia

Editor Comments

1. Comment: In your Methods section, please provide additional location information of the sampling sites, including geographic coordinates for the data set if available.

Reply: Comment accepted and done accordingly (Page 7 lines 186-189).

2. Comment: In your Methods section, please provide additional information regarding the permits you obtained for the work. Please ensure you have included the full name of the authority that approved the sampling sites access and, if no permits were required, a brief statement explaining why.

Reply: Included on page 12 lines 339-343.

3. Additional Editor Comments (if provided): Both reviewers have raised concerns regarding the fact that important information about the soils and the agricultural systems and practices have not been taken into account for the interpretation of your results, suggesting that irrigation might not be the most important factor explaining the differences in soil quality. I agree with them, so I would suggest to revise your work taking into account their comments in this direction.

Reply: Revised accordingly (pls see the track changes in the paper)

Reviewers' comments: Reviewer's Responses to Questions

Comments to the Author

1. Is the manuscript technically sound, and do the data support the conclusions?

Reviewer #1: Partly

Reviewer #2: Partly

Reply: Revised to make the manuscript technically fully sound, and data support the conclusions (pls see marked paper)

2. Has the statistical analysis been performed appropriately and rigorously?

Reviewer #1: Yes

Reviewer #2: No

Reply: Addressed the concerns of Reviewer #2 as shown on the track changes (particularly pls see page 10 lines 264-265; and page 11 lines 293-298)

3. Have the authors made all data underlying the findings in their manuscript fully available?

Reviewer #1: Yes

Reviewer #2: Yes

Reply: We have accepted the PLOS ONE policy

4. Is the manuscript presented in an intelligible fashion and written in standard English?

Reviewer #1: Yes

Reviewer #2: Yes

Reply: Thanks, noted.

5. Review Comments to the Author

Reviewer #1: This manuscript reports an attempt to identify key factors affecting soil quality (productivity) in a region where little is known about the interaction of soil properties and soil quality in small farmer agricultural production systems. Specific Comments are:

Comment 1. Significant additional editing of English usage and grammar are needed throughout the manuscript.

Reply: Comment accepted and incorporated accordingly as it has shown as track changes.

Comment 2. Page 3, line 79: Change “have practiced” to “have been implemented”.

Reply: Comment incorporated on page 4 line 79.

Comment 3. Page 5, line 116: Change “to deal” to ‘dealing”.

Reply: Comment accepted (page 5 line 116)

Comment 4. Page 5, line 123: Insert “have” between “that” and “generalized”.

Reply: Comment accepted and done accordingly (page 5 line 124).

Comment 5. Page 5, lines 129-132: “…cannot be said…”? English editing???

Reply: Revised accordingly (page 5 line 131-132)

Comment 6. Page 5, lines 137-142: (Objectives) Are these properties to be related to specific land use/management practices? Is this information intended to be used only in the Dura sub-catchment or also used in other areas of Ethiopia? It might make this study more valuable in the authors could indicate if this concept is a “stand alone” effort for a small part of Ethiopia or whether it is part of a larger plan or concept for improving the agricultural lands in Ethiopia. Or, is it designed to specifically for education of small farmers. Placing this study into a larger context makes this work more valuable.

Reply: Done according to the comment (page 6 lines 141-142).

Comment 7. Page 6, line 170: What is an “irrigation command area”? Is it an irrigation project area? Is it an area where irrigated agriculture predominates? Is this an area where irrigation water is available but is used at the choice of the farmer? Is part of this study designed to evaluate the impacts of irrigation on soil quality outside of just making more water available for crops? An explanation of how this fits into the overall agricultural picture in the sub-catchment would provide a clearer context for the conduct of the study.

Reply: Revised accordingly on page 7 line 172.

Comment 8. Page 12, lines 328-336: It is not clear from the description of the sites what the actual soil types and parent materials exist at each sampled site. Differenced is sand can be due to erosion, deposition of erosion sediments or original soil parent material. The high clay component for the R1 and R2 systems are probably due to being in a more level depositional or alluvial landscape and may represent the Chromic Vertisols. Because they are nearly level and higher in clay, they are most suitable for irrigated agriculture. Care must be taken in interpreting data from a study like this because the properties of the soils may not be due to land use but rather the land use may be due to soil properties. A danger of this type of study is that it may be inappropriately interpreted if it is generalized over a larger area. The nature of soils in an area usually dictates how they are used. Also, use of fertilizer doesn’t usually affect soil texture. Inherent soil properties and erosion might be responsible, however. The fertilizer might have some effect on soil structure but not on soil texture.

Reply: Revised accordingly (page 7 lines 194-201; page 13 lines 355-362)

Comment 9. Page 12, lines 337-347: See the previous comment.

Reply: Revised accordingly (page 7 lines 194-201; page 13 lines 355-362)

Comment 10. Page 15, lines 429-445: The SAR of all of these soils in <1which means that Na is not a problem in the soil of the area, The authors indicate that the higher Ex Na in the TG is due to irrigation water runoff into the TG. However, water runoff into TG may or may not be responsible for the higher Ex Na. If it is, then the authors should show a link to the Na in the irrigation water (water quality?). It could also be due to the properties of the parent material.

Reply: Explanations included on page 16 lines 469-473.

Comment 11. Page 17, lines 507-509: Irrigation may also improve microbial activity because of more soil water conditions. This is difficult to interpret from C:N ratios. The C:N ratio of these soils are in the range of 10-12 which is quite normal for most soils. The TM (lowest) and TG (highest) may be the only systems that reflect the management systems but needs and examination of C and N inputs and erosion in these systems.

Reply: Explanations included on page 18 lines 551-557 during the revision.

Comment 12. Page 19, lines 563-566: This is difficult to interpret without knowing present and past erosion history.

Reply: Since sever erosion reports indicated from poor land use and management practices, it is possible to interpret it accordingly. Additional explanation has included on page 21 lines 617-620.

Comment 13. Page 19, line 580: Change “soil prosperities” to “soil properties”.

Reply: Done accordingly on page 21 line 633

Comment 14. Table 1: The rates of fertilizer need to be discussed more clearly either in the table or in the text. Potentially, information about whether DAP and urea were supplied annually, for each crop if double cropped or only occasionally within a cropping system is important in order to interpret the N and C status of the soils. “SWC practices were used at field borders” - What does this mean?

Reply: Comment accepted and described accordingly on Table 1. 

Comment 15. Table 6: Generally in summary, the clay/CEC and SOM components explain most of the differences between the systems. Right? The first two PCA’s relate to the inherent soil properties and the last three relate to management. This again illustrates the need for knowledge of distinct soil types because these are mainly responsible for soil properties across management systems.

Reply: It could be for both the inherent soil properties and management practices. But in this study, it described that the differences in the component factors should be related to management practices and effect on erosion severity. Because even CEC and SOM are highly influenced by management practices and thereby the effects of SOM can be reflected on soil bulk density and soil N.. 

Reviewer #2: General comments

Comment: This work describes in detail the results of a survey conducted to identify contrasting agrosystems and other land uses in a small catchment in Northern Ethiopia, including irrigated and rainfed systems, a reference forest soil and eroded areas with and without reclamation measures.

In this sense, the study provides detailed information on the nine systems selected, both on terms of management and about soil physical and chemical properties of the studied upper layer (0-20 cm). It adds this valuable information to the existing knowledge on land uses in the region.

From these data, authors describe the major characteristics of the soils under these managements, and use PCA to depict those properties explaining better the observed variability, with the aim to identify useful soil quality indicators for further studies. This approach is interesting as it helps also summarizing the information obtained about the 28 soil properties studied.

From this, the study seems a good contribution to the knowledge of soil condition in the study area, and to the relationship between land uses and soil properties. From this perspective, it is therefore worth publication, although, to my view, there are some issues that would need to be addressed before acceptation:

As a major concern, the results of the different properties analyzed are studied as if the only factor possibly inducing differences was the type of soil management. This was however probably not the case, as significant differences were observed in relatively unchangeable soil properties such as texture. Accepting that some soil management conditions leading to intense erosion can result in textural changes at the topsoil (lines 330-332), the observed differences in clay content (from 73.7% in R2 to 26.3 under TM, Table 3) are very likely due to soil heterogeneity. Although the authors declare that all the plots selected were on the same type of soil (Chromic Vertisol), this is information is not clear enough to warrant similar soil characteristics in all plots. This seems therefore a study based in pseudo-replication, and this should be accounted for in the description of results and the discussion. As soil texture is determinant for many of the studied soil properties such as CEC, this heterogeneity imposes limitations to the comparisons among sites. This is acknowledged by the authors (line327) but not really considered in the discussion.

Reply: Revised accordingly with detail explanations on how management practices and erosion leads to textural variability in the same soil type (page 7 lines 194-201; page 13 lines 355-362)

Comment: In relation to the objective, although irrigation as a soil management system that can induce changes in soil condition is commented with some detail in the introduction, and included in the title, the study does not allow to draw clear conclusions on the effect of irrigation per se, as it compares irrigated and non-irrigated systems which differ in many other factors. For instance, the non irrigated systems (TM and MM) are monocultures, and the irrigated systems include intercropping and rotations, but not monoculture. Also MM and TM differ in the use of organic amendments in MM but not TM. Therefore, it seems more logical to compare the agrosystems between them (as such), but not trying to derive conclusions on the effect of irrigation as an isolated factor. My suggestion would be to remove from the title the idea that the study compares irrigated vs non-irrigated systems, and focus on the fact that it makes a comprehensive revision of the most common agrosystems and land condition (forest, gullies) found in the region.

Reply: Comment accepted and revised accordingly starting from the title. But attention was given on the two agricultural practices which are irrigation and non irrigation (rain-fed) to show the roles of such bold activities in the CLUS. 

Comment: Some of the properties studied, and in particular aggregate stability (but also SAR and porosity for instance) were not directly analyzed, but derived from others in the dataset. This needs to be handled with care when studying correlation in the PCA, as of course variables calculated from other variables would correlate with them. In particular, the formula used to calculate aggregate stability needs to be revised (line 248): as it is written, under equal SOC concentrations, this expression would result in a lower value (less stability) for a soil with more clay and silt (less sandy) than to a sandy soil with less clay + silt. This is counterintuitive and probably the cause of some unexpected results in this parameter found in Table 3.

Reply: Comment accepted and the formulas are revised to calculate with respect to each CLUS (page 10 line 265, and 277

Comment: The calculation of SOC stocks for comparing (supposedly) the same soil under different management practices needs an approach based in comparing equivalent masses (i.e., the amount of SOC in the same mass of soil for all managements considered). Although the reference used to describe the methodology for SOC storage does in fact describe this point (Ellert & Bettany, 1995), the authors did not use this approach. As differences were observed in bulk density, using the same formula provided in line 274 for two soils with the same SOC concentration but different bulk density would result in a greater stock in the one with higher bulk density. To avoid this bias, the approach of an equivalent mass comparison requires some easy calculations to correct the depth (T) considered in each case. As a summary, my recommendation is that the authors consider these aspects and run the comparative study with the new perspective and data, including PCA. I hope that these comments, which are meant to be constructive, are of use for a fruitful revision of the manuscript.

Reply: Comment accepted and revised accordingly on page 10 lines 290-291; page 11 lines 292-298

Some other minor comments are:

Introduction:

Comment: L.75-76: Irrigation can indeed increase SOC by increasing primary productivity, but contrasting results have been reported in this sense (see for instance recently daGama et al., 2019. Agronomy 2019, 9, 132; doi:10.3390/agronomy9030132).

Reply: Correct. If irrigation causes salinity related problems the practice can decrease SOC. However, in the conditions of the study area where salinity is not a problem, irrigation can increase SOC by improving biomass production. This has explained on page 3 lines 75-76.

Comment: L.103-109: This paragraph would be better in the paragraph in lines 61-62, as it contains very similar information.

Reply: Comment accepted and included on page 4 lines 92-98.

Comment: L.82. Soil organic C is a nutrient? For what organisms? For plants?

Reply: Explanation included on page 4 and lines 82-83.

Comment: L.137-142. See previous comment on the problems to compare irrigation as a factor in this study in relation to objective one. Suggest to replace by “under different cropping and other land-use systems”.

Reply: Comment accepted and revised accordingly starting from the title page and through out the paper. But there are conditions to mention irrigation and non-irrigated practices.

Materials and methods

Comment: L.162. Same comment on irrigated and non-irrigated. Why stressing this and not for instance adjacent mono and poli-culture systems?

Reply: Comment accepted and revised accordingly (Page 6 lines 164-166.

Comment: L.209-211. The reader has to rely about the homogeneity of the different areas based on this statement. However, results (Table 3) indicate that soils were different at least in one genetic characteristic such as topsoil texture... This should be considered in the discussion.

Reply: Comment accepted (Page 7 lines 196-201; page 8 lines 202-203); and page 13 lines 356-364.

Comment: L.226 no need to indicate irrigated and non-irrigated, as undisturbed soil samples were also collected from the native forest and the unreclaimed gully for instance...

Reply: Corrected page 9 line 245. 

Comment: L.247-254. See previous comment on this index for soil structural stability. If this parameter is not to be measured directly on soil samples, to my view, better formulas are available in the literature to relate texture and organic matter to aggregate stability, that consider the role of BOTH organic matter and soil clays in structure stabilization.

Reply: There may be other formulas, but the existing formula is suggested in different sources to be used to assess soil structure stability in which this is also adopted to the study area conditions. 

Comment: L.272-280. See comments above on the use of equivalent masses to compare SOC (and TN) stocks.

Reply: Comment accepted and incorporated as shown on page 11 lines 292-300.

Comment: L.285-289. Without commenting on the adequateness of ANOVA for this treatment, management systems are used here as one factor, without separating “irrigation” as an independent factor. This supports my previous observations that irrigation should not be commented in this work as a factor of study. It just makes part of some of the agrosystems studied, and it can be used to explain the results, just as organic fertilization, afforestation, tillage and the other parameters included in the management routines of the agrosystems described in Table 1.

Reply: Comment accepted and ANOVA was used a factor of the different CLUS. Irrigation and non irrigation was considered as part of the practices in the CLUS.

Results & Discussion

Comment: L.322-327. See comments above on the difference in texture. It seems unlikely (and cannot be proved with this study) that soil management is the responsible of all the differences observed in clay content among the nine studied sites.

Reply: Comment accepted (Page 7 lines 196-201; page 8 lines 202-203); and page 13 lines 356-364.

Comment: L.390-391. Why should irrigation increase the soil pH? Is there anything in irrigation water quality that suggests so? Why it would only affect R1 and not the other irrigated treatments?

Reply: Explanation included on page 15 lines 423-426.

Comment: L.421-422. What is the relationship between soil management and CEC?

Reply: Explanation is shown on page 16 lines 460-462

Comment: L.429-445. Some discussion is done here on NA and SAR. I agree that the values reported are not within the range of sodicity, but could the different values observed between treatments affect for instance soil aggregation and therefore infiltration, clogging, etc?

Reply: Explanations included on page 16 lines 471-476; and page 17 lines 488 – 491.

Comment: L.505. So we discover here that the “non-irrigated” TM treatment is water-saturated more than 6 months per year. How can this have changed soil properties other than SOC?

Reply: Explanation included on page 19 lines 553-556.

Comment: L.534-576. See comments above on the comparisons on SOC that require equivalent mass corrections.

Reply: Comment accepted and incorporated as shown on page 11 lines 292-300.

Comment: L.580 Soil prosperities? (guess properties...)

Reply: Corrected on page 21 line 636.

Comment: L. 596-623. See previous comment on covariance and correlation of variables derived mathematically from each other.

Reply: Revised accordingly to reflect each CLUS.

Conclusions

Comment. Can be revised after addressing previous comments.

Reply: Revised accordingly (pls see on page 24)

---

## [Decision Letter · Decision Letter 1]

30 Oct 2019

PONE-D-19-24034R1

Variation in soil properties under different cropping and other land-use systems in Dura catchment, Northern Ethiopia

PLOS ONE

Dear Dr. Tesfahunegn,

Thank you for submitting your manuscript to PLOS ONE. After careful consideration, we feel that it has merit but does not fully meet PLOS ONE’s publication criteria as it currently stands. Therefore, we invite you to submit a revised version of the manuscript that addresses the points raised during the review process.

We would appreciate receiving your revised manuscript by Dec 14 2019 11:59PM. To enhance the reproducibility of your results, we recommend that if applicable you deposit your laboratory protocols in protocols.io, where a protocol can be assigned its own identifier (DOI) such that it can be cited independently in the future. For instructions see: http://journals.plos.org/plosone/s/submission-guidelines#loc-laboratory-protocols

We look forward to receiving your revised manuscript.

Kind regards,

Remigio Paradelo Núñez

Academic Editor

PLOS ONE

Additional Editor Comments (if provided):

Dear authors,

Both reviewers agree that the modifications performed have improved the manuscript, but they still feel that several changes must be made before publication. In addition to their specific comments, it should be more clearly acknowledged that the soil variability observed in your study could be due to intrinsic soil heterogeneity in such a wide area, and not only to cropping system or land-use.

Reviewers' comments:

Reviewer's Responses to Questions

**Comments to the Author**

1. If the authors have adequately addressed your comments raised in a previous round of review and you feel that this manuscript is now acceptable for publication, you may indicate that here to bypass the “Comments to the Author” section, enter your conflict of interest statement in the “Confidential to Editor” section, and submit your "Accept" recommendation.

Reviewer #1: All comments have been addressed

Reviewer #2: (No Response)

2. Is the manuscript technically sound, and do the data support the conclusions?

Reviewer #1: Yes

Reviewer #2: Partly

3. Has the statistical analysis been performed appropriately and rigorously? 

Reviewer #1: Yes

Reviewer #2: No

4. Have the authors made all data underlying the findings in their manuscript fully available?

Reviewer #1: Yes

Reviewer #2: Yes

5. Is the manuscript presented in an intelligible fashion and written in standard English?

Reviewer #1: Yes

Reviewer #2: Yes

6. Review Comments to the Author

Reviewer #1: The authors have satisfied most of this reviewer’s concerns. However, a few concerns still remain:

1. Pages 19-20, lines 571-610: Change “OC:TN” ratio to “C:N” ratio. This is the most commonly used form and is understood by most soil scientists and agronomists.

2. Page 20, lines 581-591: The authors seem to place a great deal of focus on soil C:N ratios. Soil C:N ratios are a function of quality of inputs minus the outputs of soil biological systems. The C:N ratios of these soils range from 9.6 to 13.7. These values reflect the “balance” of the inputs (C and N materials) vs. the outputs (CO2 and mineral N to the crops or leaching) as mediated by soil microorganisms. These value generally lie in the normal range of 10-12 (mean=11.8) with the outliers very close to this range. The extreme of 9.6 (TM) may reflect the annual N fertilization that might be greater than the crop needs. The other extreme of 13.7 (TG) may reflect no outside fertilization and the natural biomass inputs from the grasses growing on the land. The leguminous trees may or may not have much impact on the soil C:N status. Unless there are soil samplings at or near fertilizer or plant residue applications, the C:N ratios appear to be in “balance” with each system. When C:N ratios are near normal, then minor differences in C:N ratio will have very little effect on soil productivity because other productivity factors (such as water availability) will override any fertility influenced by the soil C:N ratio.

3. Additional English editing is still needed.

This manuscript should be accepted for publication after minor revision based on the above comments. This is an interesting study that is useful for farm managers in many regions of the globe.

Reviewer #2: PONE-D-19-24034.R1

Variation in soil properties under different cropping and other land-use systems in Dura catchment, Northern Ethiopia

General comments

This manuscript is a revised version of a work I revised before. I stand to my previous comments that the study is of interest, mostly because it adds comprehensive information about soil parameters in an area where this is scarce. It also offers an overview of soil variability in the region, and in relation to cropping and land-use systems (CLUS). I appreciate the approach based in participation of farmers in the selection of CLUS and representative plots.

In my previous report, I addressed however some points of concern in relation to the approach, some methodologies, and, in particular, the impossibility of deriving conclusions related only to the effect of CLUS in the region from the sampling scheme designed and conducted in the study.

I appreciate the efforts done by the authors to modify and improve the manuscript to this respect. Still, in addition to some methodological considerations (see below), which, to my view are not still clear and could be improved, there is still in the manuscript a view that implies establishing a cause-effect relationship between CLUS and soil condition (soil parameters). Although this is very likely the case for many properties, it is still difficult to be completely assertive (as authors are) in this respect considering that only one plot per CLUS was sampled in the whole catchment area, and that no clear information on soil natural heterogeneity is provided.

For instance, authors (lines 194-201) assign the observed (very high) variability in topsoil texture to land management. The explanations provided seem reasonable, but still this is more an observational information than evidence-based data. Also, they acknowledge later (l.355) that such a heterogeneity “could influence the other textural classes and physical and chemical soil properties”. If so, the question remains open on whether it is soil characteristics which induce the particular selection of one use, or if it is the use which results in the observed status of soil parameters.

With this (and the comments below), my view is that the manuscript needs to be revised (mostly in the discussion) to account for this particular issue.

In addition, and also related to my previous comments, some methodological aspects that would need some clarification are:

- The calculation of SOC stocks using an equivalent mass approach needs, as a first step, to determine such equivalent mass. As the formulae provided are not clear to this respect, and data of bulk density and SOC and NT stocks provided in Table 5 show clear differences among CLUS (Table 3), I believe it would be easier for readers to provide the value of the equivalent mass chosen in the M&M section, when the method is explained, and then provide SOC and TN stocks data in relation to that mass.

- The pedo-transfer equation used for aggregate stability has not changed since the first version. Authors argue that it has been used in other studies and therefore validated by evidence. Since it still implies that same SOC values in a soil with less clay would result in lower aggregate stability than in a soil with more clay, I’d suggest that the authors provide at least the references they mention to support its use.

- The PCA has not changed since the original version, nor its discussion. Still, I believe some comments are needed on the use of derived properties such as aggregate stability, which generate auto-correlation based in calculations, not measured data.

7. PLOS authors have the option to publish the peer review history of their article (what does this mean?). If published, this will include your full peer review and any attached files.

Reviewer #1: No

Reviewer #2: No

---

## [Author Response · Author response to Decision Letter 1]

28 Nov 2019

Journal: PLOS ONE

PONE-D-19-24034R1

Variation in soil properties under different cropping and other land-use systems in Dura catchment, Northern Ethiopia

Editor Comments

Comment: Both reviewers agree that the modifications performed have improved the manuscript, but they still feel that several changes must be made before publication. In addition to their specific comments, it should be more clearly acknowledged that the soil variability observed in your study could be due to intrinsic soil heterogeneity in such a wide area, and not only to cropping system or land-use.

Reply: Yes, of course, that is why the PCA determinant soil parameters explain the soil quality variability by the CLUS by 87%, indicating the rest 13% could be intrinsic factors (page 24 line 709).

Reviewer's Responses to Questions

Comments to the Author

1. If the authors have adequately addressed your comments raised in a previous round of review and you feel that this manuscript is now acceptable for publication, you may indicate that here to bypass the “Comments to the Author” section, enter your conflict of interest statement in the “Confidential to Editor” section, and submit your "Accept" recommendation.

Reviewer #1: All comments have been addressed

Reviewer #2: (No Response)

Reply with reference to Reviewer #2: In this revision, your comments were well addressed. Pls see the track changes in the paper. ________________________________________

2. Is the manuscript technically sound, and do the data support the conclusions?

Reviewer #1: Yes

Reviewer #2: Partly

Reply with reference to Reviewer #2: Pls seethe revised conclusion on page 26.________________________________________

3. Has the statistical analysis been performed appropriately and rigorously?

Reviewer #1: Yes

Reviewer #2: No

Reply with reference to Reviewer #2: Revised paper with due attention to the calculation of soil mass, equivalent soil mass and the OC and TN stocks as indicated on page 11 lines 299-318; page 12 lines 319-342; and Table 5.________________________________________

4. Have the authors made all data underlying the findings in their manuscript fully available?

Reviewer #1: Yes

Reviewer #2: Yes

5. Is the manuscript presented in an intelligible fashion and written in standard English?

Reviewer #1: Yes

Reviewer #2: Yes

6. Review Comments to the Author

Reviewer #1: The authors have satisfied most of this reviewer’s concerns. However, a few concerns still remain:

Comment: 1. Pages 19-20, lines 571-610: Change “OC:TN” ratio to “C:N” ratio. This is the most commonly used form and is understood by most soil scientists and agronomists.

Reply: Comment accepted and done accordingly (Page 11 lines 292-297 ).

Comment: 2. Page 20, lines 581-591: The authors seem to place a great deal of focus on soil C:N ratios. Soil C:N ratios are a function of quality of inputs minus the outputs of soil biological systems. The C:N ratios of these soils range from 9.6 to 13.7. These values reflect the “balance” of the inputs (C and N materials) vs. the outputs (CO2 and mineral N to the crops or leaching) as mediated by soil microorganisms. These value generally lie in the normal range of 10-12 (mean=11.8) with the outliers very close to this range. The extreme of 9.6 (TM) may reflect the annual N fertilization that might be greater than the crop needs. The other extreme of 13.7 (TG) may reflect no outside fertilization and the natural biomass inputs from the grasses growing on the land. The leguminous trees may or may not have much impact on the soil C:N status. Unless there are soil samplings at or near fertilizer or plant residue applications, the C:N ratios appear to be in “balance” with each system. When C:N ratios are near normal, then minor differences in C:N ratio will have very little effect on soil productivity because other productivity factors (such as water availability) will override any fertility influenced by the soil C:N ratio.

Reply: Revision was made considering the comment above (Page 20 line 594; page 21 lines 612-615 ). However, with regard to the C:N ratios range (10-12) commented as the ratios lie in the normal range, but explanations are available on page 21 lines 612-615 as it indicates that C:N ratio less than 10 is good (normal) soil condition.

Comment: 3. Additional English editing is still needed.

Reply: Editing was done accordingly (see throughout the paper for marked changes)

Decision of reviewer #1: This manuscript should be accepted for publication after minor revision based on the above comments. This is an interesting study that is useful for farm managers in many regions of the globe.

Reviewer #2: PONE-D-19-24034.R1

Variation in soil properties under different cropping and other land-use systems in Dura catchment, Northern Ethiopia

General comments

This manuscript is a revised version of a work I revised before. I stand to my previous comments that the study is of interest, mostly because it adds comprehensive information about soil parameters in an area where this is scarce. It also offers an overview of soil variability in the region, and in relation to cropping and land-use systems (CLUS). I appreciate the approach based in participation of farmers in the selection of CLUS and representative plots.

In my previous report, I addressed however some points of concern in relation to the approach, some methodologies, and, in particular, the impossibility of deriving conclusions related only to the effect of CLUS in the region from the sampling scheme designed and conducted in the study.

I appreciate the efforts done by the authors to modify and improve the manuscript to this respect. Still, in addition to some methodological considerations (see below), which, to my view are not still clear and could be improved, there is still in the manuscript a view that implies establishing a cause-effect relationship between CLUS and soil condition (soil parameters). Although this is very likely the case for many properties, it is still difficult to be completely assertive (as authors are) in this respect considering that only one plot per CLUS was sampled in the whole catchment area, and that no clear information on soil natural heterogeneity is provided.

Reply: Three plots per CLUS was used as this is indicated on page 10 lines 233-235. 

Comment: For instance, authors (lines 194-201) assign the observed (very high) variability in topsoil texture to land management. The explanations provided seem reasonable, but still this is more an observational information than evidence-based data. Also, they acknowledge later (l.355) that such a heterogeneity “could influence the other textural classes and physical and chemical soil properties”. If so, the question remains open on whether it is soil characteristics which induce the particular selection of one use, or if it is the use which results in the observed status of soil parameters.

With this (and the comments below), my view is that the manuscript needs to be revised (mostly in the discussion) to account for this particular issue.

Reply-a: With regard to the observation information which is commented as it is not evidence-based data, it is quiet write to have evidence-based data. However, to have such data it demands additional research which is not in line with the purpose of the present paper. Rather our explanation was relied on the logical information collected from the farmers discussion and field observation that states that there is long-term variability in management practices and cropping systems in which such variation could lead to differences in soil properties including soil texture (Page 7 lines 173-197). In addition, it is indicated that the soil properties variability is mainly related to the CLUS (87%), but the rest 13% may be intrinsic factor (Page 24 lines 709-710 ).

Reply-b: With regard to the comment “Also, they acknowledge later (l.355) that such a heterogeneity “could influence the other textural classes and physical and chemical soil properties”. If so, the question remains open on whether it is soil characteristics which induce the particular selection of one use, or if it is the use which results in the observed status of soil parameters.”

The reply to this comment is that “the sentences …influence the other textural classes and physical and chemical soil properties” is deleted during this revision in order to reduce confusion by the readers so that to make clear the cause for the soil properties variability in the study area is related to the different CLUS and their corresponding soil mgt practices (page 14 lines 397-398). 

Specific Comment: In addition, and also related to my previous comments, some methodological aspects that would need some clarification are:

Comment 1- The calculation of SOC stocks using an equivalent mass approach needs, as a first step, to determine such equivalent mass. As the formulae provided are not clear to this respect, and data of bulk density and SOC and NT stocks provided in Table 5 show clear differences among CLUS (Table 3), I believe it would be easier for readers to provide the value of the equivalent mass chosen in the M&M section, when the method is explained, and then provide SOC and TN stocks data in relation to that mass.

Reply: Comment accepted and revised accordingly (Page 11 lines 300-319; page 12 lines 320-343) and also Table 5.

Comment 2- The pedo-transfer equation used for aggregate stability has not changed since the first version. Authors argue that it has been used in other studies and therefore validated by evidence. Since it still implies that same SOC values in a soil with less clay would result in lower aggregate stability than in a soil with more clay, I’d suggest that the authors provide at least the references they mention to support its use.

Reply: Revision Done accordingly (page 10 line 277-282). Plus see page 16 lines 455-457.

Comment 3- The PCA has not changed since the original version, nor its discussion. Still, I believe some comments are needed on the use of derived properties such as aggregate stability, which generate auto-correlation based in calculations, not measured data.

Reply: Yes of course, on the concern of auto-correlation, but it was handle by partial correlation analysis and also excluded from the PC considering the correlation and loadings as indicated on page 24 lines 701-705; 722-725; page 25 lines 737-750)

I sincerely thank you the editor and the anonymous reviewers for the constructive comments and suggestion. 

The End!

---

## [Editor Report · Decision Letter 2]

4 Dec 2019

Variation in soil properties under different cropping and other land-use systems in Dura catchment, Northern Ethiopia

PONE-D-19-24034R2

Dear Dr. Tesfahunegn,

We are pleased to inform you that your manuscript has been judged scientifically suitable for publication and will be formally accepted for publication once it complies with all outstanding technical requirements.

With kind regards,

Remigio Paradelo Núñez

Academic Editor

PLOS ONE
---

## [Editor Report · Acceptance letter]

10 Dec 2019

PONE-D-19-24034R2 

Variation in soil properties under different cropping and other land-use systems in Dura catchment, Northern Ethiopia 

Dear Dr. Tesfahunegn:

I am pleased to inform you that your manuscript has been deemed suitable for publication in PLOS ONE. Congratulations! Your manuscript is now with our production department. 

With kind regards,

on behalf of

Dr. Remigio Paradelo Núñez 

Academic Editor

PLOS ONE